# Biomarker characterization of the North Water Polynya, Baffin Bay: Implications for local sea ice and temperature proxies

David J. Harning[1], Brooke Holman[1], Lineke Woelders[1], Anne E. Jennings[1], Julio Sepúlveda[1, 2]

[1]Institute of Arctic and Alpine Research, University of Colorado, Boulder, USA

[2]Department of Geological Sciences, University of Colorado, Boulder, USA

*Correspondence to*: David J. Harning (david.harning@colorado.edu)

**Abstract.** The North Water Polynya (NOW, Inuktitut: *Sarvarjuaq*, Kalaallisut: *Pikialasorsuaq*), Baffin Bay, is the largest polynya and one of the most productive regions in the Arctic. This area of thin to absent sea ice is a critical moisture source for local ice sheet sustenance and coupled with the inflow of nutrient-rich Arctic Surface Water, supports a diverse community

of Arctic fauna and indigenous people. Although paleoceanographic records provide important insight into the NOW's past behavior, it is critical that we better understand the modern functionality of paleoceanographic proxies. In this study, we analyzed lipid biomarkers, including algal highly-branched isoprenoids and sterols for sea ice extent and pelagic productivity, and archaeal GDGTs for ocean temperature, in a set of modern surface sediment samples from within and around the NOW. In conjunction with previously published datasets, our results show that all highly-branched isoprenoids exhibit strong

correlations with each other and not with sterols, which suggests a spring/autumn sea ice diatom source for all HBIs rather than a combination of sea ice and open water diatoms as seen elsewhere in the Arctic. Sterols are also highly concentrated in the NOW and exhibit statistically higher concentrations here compared to sites south of the NOW, consistent with the order of magnitude higher primary productivity observed within the NOW relative to surrounding waters in spring/summer months. Finally, our local temperature calibrations for GDGTs and OH-GDGTs reduce the uncertainty present in global temperature

calibrations, but also identify some additional variables that may be important in controlling their local distribution, such as nitrate availability and dissolved oxygen. Collectively, our analyses provide new insight into the utility of these lipid biomarker proxies in high-latitude settings and will help provide a refined perspective on the past development of the NOW with their application in downcore reconstructions.

## 1 Introduction

Arctic and Antarctic polynyas are key sites for deep water formation (Kuhlbrodt et al., 2017), ice sheet moisture (Smith et al., 2010), and enhanced productivity that can sequester atmospheric $CO_2$ (Arrigo and van Dijken, 2004). As global temperatures continue to climb, further reductions in sea ice are projected along Arctic coastlines (Barnhart et al., 2015; Onarheim et al., 2018), calling the future status of polynyas into question. Polynyas will cease to exist where seasonal sea ice vanishes or transitions to marginal sea ice zones, which will both result in cascading negative effects on regional and global environments

(Meredith et al., 2019; Moore et al., 2021). One way to alleviate some uncertainty about the future status of polynyas is by reconstructing changes in sea ice extent and productivity in the recent geologic past to understand how polynyas have responded to past climate change. In this light, the Holocene instability of coastal polynyas has recently been shown in the Barents Sea (Knies et al., 2018), East Greenland (Syring et al., 2020) and northern Baffin Bay (Jackson et al., 2021; Ribeiro et al., 2021) using lipid biomarker climate proxies, such as $IP_{25}$ (sea ice) and sterols (open water productivity), along with other

proxies such as microfossils and bulk geochemistry. Improved understanding of the modern distributions of biomarkers and the influencing environmental factors within dynamic Arctic regions is critical for supporting down-core interpretations (e.g., Smik and Belt, 2017; Belt et al., 2019; Kolling et al., 2020).

Here, we focus on the North Water Polynya (NOW, Inuktitut: *Sarvarjuaq*, Kalaallisut: *Pikialasorsuaq*), which is the largest polynya (85,000 km$^2$) and one of the most productive regions in the Arctic (Barber et al., 2001; Klein et al., 2002). The

latent heat NOW forms when an ice bridge consolidates at the head of Smith Sound, which restricts the passage of Arctic ice floes but allows throughflow of nutrient-rich Arctic Surface Water (ASW) into Baffin Bay through Nares Strait (Fig. 1, Melling et al., 2001). This throughflow and open water fuels high local productivity, which supports a diverse community of Arctic fauna, including higher-trophic level seabirds and marine mammals (Stirling, 1980; Tremblay et al., 2002a). Due to the rich biodiversity, the region has also supported intermittent human occupation for at least 4400 years (Grønnow and Sørensen,

2006; Raghavan et al., 2014), with modern Inuit inhabitants continuing to rely on the NOW for their food security and subsistence economy today (Hastrup et al., 2018). Given that stable ice arches fail to form as reliably as they once did (Vincent, 2019; Moore et al., 2021), the NOW is becoming geographically and seasonally less defined (Ryan and Münchow, 2017).

In this study, we aim to characterize the NOW through the distribution of lipid biomarkers archived in marine seafloor surface sediments that encompass its modern area in Baffin Bay using both new and previously published data (Kolling et al.,

2020). We focus on different lipid classes that inform us about seasonal sea ice extent, surface productivity, and ocean temperature. Our assessment of these biomarker proxies against modern instrumental data (i.e., satellite-derived sea ice extent and *in situ* environmental datasets) provides a key baseline for interpreting the presence and extent of the NOW in the geologic past (Georgiadis et al., 2020; Jackson et al., 2021). On a broader scale, our work is also critical for the community's general understanding of these lipids' environmental relationships at high northern latitudes where some proxy datasets are currently

sparse (e.g., Tierney and Tingley, 2014, 2018). For example, existing biomarker temperature calibrations are often global in scale and feature high uncertainties at the low end of the temperature spectrum (Kim et al., 2010, 2012). However, uncertainty can be substantially reduced by filling in high latitude regions and isolating their distinct characteristics with local calibrations (Tierney and Tingley, 2014, 2018; Harning et al., 2019). While some of our biomarker datasets are limited in size, they provide an important first step for the continued refinement of these proxies at high latitudes.

**2. Study Area**

Ocean circulation in Baffin Bay is cyclonic, involving the northward flowing West Greenland Current (WGC) and the southward flowing Baffin Current (BC) (Fig. 1). The warm and saline WGC carries a mixture of Atlantic Water from the

Irminger Current and Polar Water from the East Greenland Current, whereas the BC is comprised of low salinity Arctic surface water (ASW) that enters Baffin Bay from the Arctic Ocean through the Canadian Arctic Archipelago (CAA) channels. The ASW is modified by mixing with terrestrial-derived freshwater and by sea-ice processes en route to Baffin Bay (Tang et al., 2004; Münchow et al., 2006, 2015, Azetsu-Scott et al., 2010). The present-day depths of the CAA channels govern the composition of inflowing ASW (Jones et al., 2003); Nares Strait has a sill depth of 220 m that allows passage of both the Polar Mixed Layer (containing high-nutrient Pacific Water from Bering Strait) and some of the halocline layer that has been mixed with the underlying Atlantic layer of the Arctic Ocean (Azetsu-Scott et al., 2010). Lancaster and Jones Sounds have shallower sill depths that exclude all but the most carbonate-undersaturated Polar Mixed Layer (Azetsu-Scott et al., 2010). These Arctic outflows join the BC and form the upper 100 to 300 m of surface water in Baffin Bay, except where the WGC dominates in the southeast (Tang et al., 2004).

Sea ice covers nearly all of Baffin Bay in winter, except in the southeast due to the warmth and salinity of the WGC (Fig. 1). Sea ice begins to form in September and reaches maximum coverage in March and is thickest along the Baffin Island coast where the ASW flow is concentrated (Fig. 1, Tang et al., 2004). In contrast, the NOW has low concentrations of thin sea ice, even during winter months. Consolidation of an ice arch at the head of Smith Sound initiates the formation of the polynya, which is further stimulated by northerly winds and currents that remove newly formed sea ice (Ingram et al., 2002; Bi et al., 2019), and sensible heat from WGC upwelling on the Greenland side (Melling et al., 2001; Ingram et al., 2002). Baffin Bay sea ice concentrations decrease between April and August; beginning in the NOW region before propagating southward and creating a generally ice-free Baffin Bay by June (Bi et al., 2019). The Pacific Water, a major component of the ASW, has twice the nitrogen and phosphorus and seven times the silica of Atlantic Water (Jones et al., 2003). The high nutrient content of incoming ASW, along with higher light levels and stratification in the NOW, fuels high seasonal phytoplankton productivity (Lewis et al., 1996; Ingram et al., 2002; Tremblay et al., 2002a). Productivity is an order of magnitude higher in the NOW than in adjacent areas of Baffin Bay, making it one of the most important areas for new production in the Arctic (Tremblay et al., 2002a).

## 3. Background on lipid biomarkers

### 3.1. HBIs

Highly branched isoprenoids (HBIs) are unsaturated hydrocarbons (Fig. S1) biosynthesized by a narrow range of marine diatoms (see review by Belt, 2018). The mono-unsaturated HBI termed IP$_{25}$, first discovered in Canadian Arctic sea ice (Belt et al., 2007), has developed into an important proxy for seasonal sea ice due to its production during spring blooms (Brown et al., 2011; Limoges et al., 2018; Amiraux et al., 2019) of Arctic sea ice diatoms (Brown et al., 2014). The di-unsaturated HBI II likely has an Arctic sea ice diatom source as well based on distinctly heavy stable carbon isotopic composition, in addition to similar concentration profiles to IP$_{25}$ across Arctic marine surface sediment (Belt et al., 2008; Cabedo-Sanz et al., 2013; Brown et al., 2014; Limoges et al., 2018). IP$_{25}$ and HBI II below the limit of detection have often been interpreted to reflect either a lack of seasonal sea ice cover or permanent and thick sea ice that blocks sunlight penetration needed for sea ice diatom

photosynthesis. However, this is currently only an assumption, and no targeted investigations aimed at clarifying the production/deposition of IP25 under perennial sea ice cover have been carried out (Belt, 2018).

Other HBIs, such as the tri-unsaturated isomers HBI III and IV, have been attributed to biosynthesis by open-water phytoplankton (Belt et al., 2000, 2008, 2015, 2017; Rowland et al., 2001). The PIP25 index has been used to differentiate between open water and thick sea ice conditions inferred from IP25 and HBI II in the Arctic, and develop semi-quantitative sea ice reconstructions (Cabedo-Sanz et al., 2013; Smik et al., 2016; Köseoğlu et al., 2018):

$$PIP_{25} = \frac{IP_{25}}{IP_{25} + (phytoplankton\ biomarker\ x\ c)}, \tag{1}$$

where the balance factor c is a ratio of mean IP25 and mean phytoplankton biomarker concentrations (e.g., HBI III). Certain sterols (see following paragraph) can also be used as the phytoplankton biomarker in lieu of HBI III (e.g., Müller et al., 2011). Although in some regions high concentrations of HBI III have also been associated with highly productive marginal ice zones (Barents Sea, Belt et al., 2019), marine fronts (North Iceland Shelf, Harning et al., 2021) and sea ice (Amiraux et al., 2019, 2020; Koch et al., 2020) that may complicate PIP25-derived indices, recent compilations of Arctic surface sediments show that PIP25-based indices broadly correlate with spring and autumn sea ice concentrations (Xiao et al., 2015; Kolling et al., 2020).

## 3.2. Sterols

Sterols are ubiquitous components in eukaryotic organisms (Fig. S2, Volkman,1986), and similar to HBI III and IV, have become common complementary biomarkers in IP25 and PIP25 datasets. Although these biomarkers have often been attributed to specific sources, such as pelagic phytoplankton (brassicasterol, e.g., Navarro-Rodriguez et al., 2013), dinoflagellates (dinosterol, e.g., Boon et al., 1979) and vascular plants (campesterol and β-sitosterol, e.g., Huang and Meinschein, 1976), they are now known to derive from multiple sources that complicate their source specificity. For example, brassicasterol and dinosterol are also produced by sea ice algae (Nichols et al., 1990; Belt et al., 2013, 2018) and pennate diatoms (e.g., Volkman et al., 1993; Rampen et al., 2010), and campesterol and β-sitosterol can be produced by diatoms as well by vascular plants (Belt et al., 2013, 2018; Rampen et al., 2010). Hence, the abundance of these sterols within marine sedimentary records is more broadly reflective of general marine productivity (e.g., Köseoğlu et al., 2019). In the CAA where terrestrial biomass is low (Gould et al., 2003), we further assume that the contribution of terrestrial-derived campesterol and β-sitosterol is minimal compared to that produced in the ocean.

## 3.3. GDGTs

Isoprenoid glycerol dibiphytanyl glycerol tetraethers (GDGTs) are cell membrane-spanning lipids biosynthesized by archaea (Fig. S3, Pearson and Ingalls, 2013), including ammonia oxidizing Thaumarchaeota (Schouten et al., 2002; Könneke et al., 2005; Pitcher et al., 2011; Besseling et al., 2020). Thaumarchaeota can modify the number of cyclopentane moieties (cyclization) in GDGTs in response to *in situ* temperature variability, a process known as homeoviscous adaptation (e.g., Elling et al., 2015). Thus, correlations are found between the degree of GDGT cyclization in global surface sediment datasets and

upper ocean temperature. These degrees of cyclization are commonly reported with various versions of the tetraether index of

tetraethers consisting of 86 carbons (TEX$_{86}$) index (Schouten et al., 2002, 2013; Kim et al., 2010, 2012).

Evidence from a latitudinal transect in the western Atlantic Ocean demonstrates that GDGTs are most likely produced and exported to the seafloor from 80–250 m water depth (Hurley et al., 2018), which compares well to archaea abundance maxima at 200 m water depth in the Pacific Ocean (Karner et al., 2001). Considering that Thaumarchaeota are chemolithoautotrophs that perform ammonia oxidation (conversion of ammonia to nitrite), they are typically more abundant

around the primary nitrite maximum near the base of the photic zone (Church et al., 2010; Francis et al., 2005; Hurley et al., 2018) and are most productive when there is minimized phytoplanktic competition over ammonia (Schouten et al., 2013). In the higher latitudes, the latter occurs during the less productive dark winter months when photosynthesis for sea surface species is inhibited, which may explain the seasonal winter temperature bias of GDGTs observed in this latitudinal band (Herfort et al., 2006; Rueda et al., 2009; Rodrigo-Gámiz et al., 2015; Harning et al., 2019). Although the temperature relationship of

TEX$_{86}$-based indices deviates from the linear global temperature calibrations and features higher uncertainty at the cold end of the spectrum (Schouten et al., 2002; Kim et al., 2010, 2012), regional calibrations have proven useful for reducing estimate uncertainty (Harning et al., 2019) that may be at least partially attributed to changes in community composition (e.g., Elling et al., 2017). Moreover, new indices based on hydroxylated isoprenoid GDGTs (Fig. S3, OH-GDGTs, e.g., RI-OH and RI-OH' indices, Lü et al., 2015) also produced by planktic Thaumarchaeota (Elling et al., 2017; Bale et al., 2019) have been suggested

to improve upon TEX$_{86}$-based proxies in the polar oceans (Fietz et al., 2013, 2020; Huguet et al., 2013). This is because the addition of one hydroxyl group may further reduce membrane rigidity at lower temperatures (Huguet et al., 2017). In addition to temperature, recent studies have shown that several other environmental and geochemical factors can influence the degree of GDGT cyclization, such as growth phase (Elling et al., 2014), ammonia oxidation rates (Hurley et al., 2016), and oxygen concentrations (Qin et al., 2015), but not likely salinity (Wuchter et al., 2004, 2005; Elling et al., 2015). Although the effects

of these environmental parameters on OH-GDGT cyclization have not been rigorously tested, emerging evidence suggests that salinity, sea ice, seasonality, and terrestrial input may be complicating factors in some oceanic settings (Fietz et al., 2013; Kang et al., 2017; Lü et al., 2019; Park et al., 2019; Wei et al., 2020; Sinninghe Damsté et al., 2022).

## 4. Materials and Methods

### 4.1. Marine surface sediment samples

We analyzed marine surface sediment samples (*n*=13) collected from box core (*n*=12) and trigger core tops (*n*=1) collected during the 2008 CSS *Hudson* and 2015 and 2017 CSS *Amundsen* research cruises, and range in water depths from 267 to 2373 m bsl (Table 1). As each sample integrates the upper 1 to 2 centimeters of sediment, they likely reflect the several most recent decades or centuries of time depending upon site-specific sedimentation rates and mixing by bioturbation. Therefore, each sample reflects time averages of biomarker production in and out of the polynya. Marine surface sediment samples collected

on these cruises as well as other over the past decades have previously been analyzed for HBIs and select sterols (i.e., brassicasterol and dinosterol) as part of larger pan-Arctic datasets (Stoynova et al., 2013; Kolling et al., 2020). To generate

larger datasets where possible, we compare our new dataset ($n$=13) with a subset of samples ($n$=70) previously published by Kolling et al. (2020). To avoid environmental settings that may be disproportionately impacted by terrestrial and/or glacier runoff, we exclude samples collected from within fjords (e.g., Kangiqtugaapik, formerly Clyde Inlet) and bays (e.g., Disko Bugt). Moreover, while the International Hydrographic Organization defines the southern limit of Baffin Bay as 70 ºN (Limits of Oceans and Seas, 1953), we extend our range into Davis Strait to 67 ºN to provide greater breadth for the environmental gradients.

## 4.2. Bulk geochemistry

At the University of Colorado Boulder's Earth Systems Stable Isotope Laboratory, freeze dried and decalcified marine surface sediment subsamples ($n$=9, Table 1) were analyzed for bulk elemental (%$CaCO_3$, %TC, %TN) and stable isotope ($\delta^{13}C$ relative and $\delta^{15}N$) geochemistry on a Thermo Delta V elemental analyzer (EA) interfaced to an isotope ratio mass spectrometer (IRMS), and computed for elemental C/N values. $\delta^{13}C$ values are expressed relative to Vienna Pee Dee Belemnite (VPDB) and $\delta^{15}N$ relative to Air. Only 9 of the total 13 samples were analyzed for bulk geochemistry as there was not enough mass in the remaining 4 for additional lipid biomarker analyses (see following section).

## 4.3. Lipid biomarkers

At the University of Colorado Boulder's Organic Geochemistry Laboratory, freeze dried marine surface sediment subsamples (~1-4 g, $n$=13) were extracted two times on a Dionex accelerated solvent extractor (ASE 200) using dichloromethane (DCM):methanol (9:1, v/v) at 100 °C and 2,000 psi. A 25 percent aliquot of the total lipid extract (TLE) was taken for GDGT analysis. The remaining 75 percent of TLE was then separated into five fractions (F1-F5) using silica column chromatography, after elution with ½ dead volume (DV) hexane (F1), 2 DV hexane:DCM (8:2, v/v) (F2), 2 DV DCM (F3), 2 DV DCM:ethyl acetate (EtOAc) (1:1, v/v) (F4) and 2 DV EtOAc (F5). Each of these extractions contained 500 ng of the following internal standards: 3-methylheneicosane (F1), p-Terphenyl-d14 (F2), docosanoic acid (F3), 1-nonadecanol (F4), 2-Me octadecanoic acid (F5).

From F1, we focus on highly branched isoprenoid (HBI) IP$_{25}$ (C$_{25:1}$), HBI II (C$_{25:2}$), HBI III (C$_{25:3}$) and HBI IV (C$_{25:3}$) biomarkers. HBIs were analyzed via gas chromatography-mass spectrometry (GC-MS) on a Thermo Trace 1310 Gas Chromatograph interfaced to a TSQ Evo 8000 triple quadrupole mass spectrometer and fitted with an Agilent DB-1MS GC column (60 m x 0.25 mm x 0.25 μm) following modified methods and operating conditions of Belt et al. (2012). We used an ion source temperature of 250 ºC rather than 300 ºC to prevent excessive fragmentation during ionization, which yields larger molecular ions that facilitated compound identification both in full scan (FS) and selected reaction monitoring (SRM) modes (e.g., Boudinot et al., 2020). The identification and quantification of IP$_{25}$ ($m/z$ 350, Belt et al., 2007), HBI II ($m/z$ 348, Belt et al., 2007), and HBI III and IV ($m/z$ 346, Belt et al., 2000) was based on their respective mass spectra compared to that of the internal standard (3-methylheneicosane, $m/z$ 310.6). To account for the varying response factors of different lipid classes and to make our sample set comparable with other published HBI datasets, we corrected the concentrations based on our internal

standard (3-methylheneicosane) by the response factor of a 5-point external HBI dilution series comprised of 7-hexylnonadecane (7-HND) and 9-octylheptadec-8-ene (9-OHD) (Fig. S4, Belt et al., 2012).

From F4, we focus on a series of diagnostic sterols, namely brassicasterol (24-Methylcholesta-5,22E-dien-3β-ol), dinosterol (4α,23,24-Trimethyl-5α-cholesta-22E-en-3β-ol), campesterol (24-Methylcholesta-5-en-3β-ol) and β-sitosterol (24-Ethylcholesta-5-en-3β-ol). Before analyses, each sample was derivatized with N,O-bis(trimethylsilyl)trifluoroacetamide (BSTFA; 25 μl) and pyridine (catalyst, 25 μl at 70 °C for 20 min.). Sterols were analyzed using the same GC-MS system described above but with an Agilent DB-5MS GC column (60 m x 0.25 mm x 0.25 μm) and under the following operating conditions: initial temperature of 80 °C (held for 2 min.), ramp 20 °C/min. (2.5 min.), ramp 5 °C/min. (68 min., held for 30 min.). Mass spectrometric analyses were carried out in full scan (FS) and selected reaction monitoring (SRM) modes. Individual sterols were identified based on their respective mass spectra (Boon et al., 1979) and then quantified by comparing their response factor to that of the internal standard (1-nonadecanol, $m/z$ 284.5). To account for the different response factor of sterols and to make our sample set directly comparable with Kolling et al. (2020), we corrected the concentrations based on our internal standard (1-nonadecanol) by the response factor of a 5-point external dilution series of cholesterol (Fig. S4).

For GDGTs, we focus on isoprenoid and hydroxylated isoprenoid GDGTs. A 25 percent aliquot of dry TLE samples was resuspended in hexane:isopropanol (99:1, v/v), sonicated, vortexed, and then filtered using a 0.45 μm polytetrafluoroethylene (PTFE) syringe filter. Prior to analysis samples were spiked with 10 ng of the $C_{46}$ GDGT internal standard (Huguet et al., 2006). GDGTs were identified and quantified via high performance liquid chromatography-mass spectrometry (HPLC-MS) following modified methods of Hopmans et al. (2016) on a Thermo Scientific Ultimate 3000 HPLC interfaced to a Q Exactive Focus Orbitrap-Quadrupole MS (Harning et al., 2019). GDGTs were identified based on their characteristic masses and elution patterns. For isoprenoid GDGTs, we explore the original $TEX_{86}$ index (Schouten et al., 2002) and the more recent $TEX_{86}^L$ index, which is a modification of the former for temperatures <15 °C (Kim et al., 2010, 2012), to reflect relative changes in temperature:

$$TEX_{86} = \frac{[GDGT-2]+[GDGT-3]+[cren.\prime]}{[GDGT-1]+[GDGT-2]+[GDGT-3]+[cren.\prime]}, \tag{2}$$

$$TEX_{86}^L = \log\left(\frac{[GDGT-2]}{[GDGT-1]+[GDGT-2]+[GDGT-3]}\right), \tag{3}$$

To evaluate the degree of influence from non-thermal factors on GDGT-derived indices, we compute the Ring Index for each sample (RI, Zhang et al., 2016):

$$RI = 0x[GDGT-0] + 1x[GDGT-1] + 2x[GDGT-2] + 3x[GDGT-3] + 4x[cren.] + 4x[cren.\prime], \tag{4}$$

and compare with the global core tope polynomial regression for $TEX_{86}$ values (Zhang et al., 2016):

$$RI_{TEX} = -0.77(\pm0.38)xTEX_{86} + 3.32(\pm0.34)x(TEX_{86})^2 + 1.59(\pm0.10), \tag{5}$$

Finally, for hydroxylated isoprenoid GDGTs, we explore two different relative temperature indices, RI-OH and RI-OH' developed for regions over and under 15 °C, respectively (Lü et al., 2015):

$$RI - OH = \frac{[OH-GDGT-1]+2x[OH-GDGT-2]}{[OH-GDGT-1]+[OH-GDGT-2]}, \tag{6}$$

$$RI - OH' = \frac{[OH-GDGT-1]+2x[OH-GDGT-2]}{[OH-GDGT-0]+[OH-GDGT-1]+[OH-GDGT-2]}, \tag{7}$$

### 4.4. Statistical analyses and WOA18 instrumental datasets

For productivity biomarkers (HBIs and sterols), we computed Pearson correlation matrices and p-values to examine their statistical relationship with one another. To determine if sites within the NOW and outside the NOW are statistically different in terms of the all studied biomarkers (HBIs, sterols, GDGTs and OH-GDGTs), we performed t-tests and computed

corresponding p-values. To assess and calibrate our GDGT-based proxies against modern climatological fields, we use World Ocean Atlas 2018 (WOA18) decadal mean datasets from 2007 to 2017 CE. We compare GDGTs against various depth integrations (0-200 mbsl) of the following variables: temperature (Locarnini et al., 2018), salinity (Zweng et al., 2018), dissolved oxygen (Garcia et al., 2018a), and nitrate (Garcia et al., 2018b) (Fig. 2). For each of these variables, we compared proxy data against the mean annual values as well as mean seasonal values where complete datasets were available. Since

winter and spring seasonal data were either fragmentary or not available, the mean annual datasets used in this study likely represent the ice-free season, integrating summer and shoulder season months in spring and autumn. All GDGT and OH-GDGT regressions are tested for significance using t-tests, and in the case of all statistical analyses, significant results are defined by those with p-values <0.05.

### 5. Results

### 5.1. Bulk geochemistry

Although only 9 out of the 13 total surface sediment samples were analyzed for bulk geochemistry, the 9 that were analyzed represent the full spatial range of our total 13 samples in Baffin Bay (Fig. 1). Bi-plots of $\delta^{13}C$ and C/N tightly cluster within the typical range of marine algae (Fig. 3a, Meyers, 1994). %TC ranges from 0.43 to 2.88 (1.60 ± 0.84, see Supplemental Data) and are generally higher in the NOW compared to non-NOW sites (Fig. 3b).

### 5.2. HBIs

HBIs are present above the detection limit in all sediment samples (*n*=13). IP$_{25}$ is the most abundant HBI in the dataset, followed by HBI II, HBI III, and HBI IV (Fig. 4a). HBIs are generally more abundant at NOW sites compared to sites outside this region, although t-test results show that only HBI II and HBI IV are statistically different between NOW and non-NOW

sites when normalized to sediment mass (g, Figs. 4a). When normalized to g TOC, the sample set is reduced to 9, and there is no statistically significant difference between NOW and non-NOW sites for any HBI (Fig. S5a). Based on Pearson correlation analysis, each of the HBIs strongly correlates with the others (>0.74, Fig. 5), which supports the similar spatial distribution of the four HBIs throughout Baffin Bay (Figs. S6-7).

### 5.3. Sterols

Sterols are present above the detection limit in all sediment samples ($n$=13). ß-sitosterol is the most abundant sterol in the dataset, followed by campesterol, brassicasterol, and dinosterol (Fig. 4c). Sterols are more abundant at NOW sites compared to sites outside this region, which t-test results show is significant in all cases when normalized to g sed (Fig. 4c). While normalizing concentrations to g TOC reduces the sample set to 9, NOW and non-NOW sites remain significantly different for all sterols (Fig. S5c). Based on Pearson correlation analysis, the sterols all strongly correlate with each other (>0.72), apart from dinosterol and ß-sitosterol whose correlation is insignificant (Fig. 5). The strong correlations between sterols supports their similar spatial distribution throughout Baffin Bay (Figs. S8-9).

### 5.4. PIP$_{25}$

We calculated new Baffin Bay balance factors (c) for P$_{III}$IP$_{25}$ (3.19), P$_{IV}$IP$_{25}$ (1.02), P$_B$IP$_{25}$ (0.54), and P$_D$IP$_{25}$ (0.63) based on a combination of the 70 previously published samples from Baffin Bay (Kolling et al., 2020) and this study's sample set to reflect the balance factors for the combined datasets. Although HBI IV was quantified by Kolling et al. (2020), we note that the authors did not explore its potential as part of a PIP$_{25}$ index. For all four PIP$_{25}$ indices in our dataset ($n$=13), we observe slightly higher mean values in non-NOW sites compared to NOW sites, although it is only statistically significant for the P$_D$IP$_{25}$ index (Fig. 4e).

### 5.5. GDGTs and OH-GDGTs

GDGTs and OH-GDGTs are present above the detection limit in all sediment samples ($n$=13). In terms of RI values, all samples plot below the global core top polynomial regression for TEX$_{86}$ (Fig. 6a). In terms of total fractional abundance, GDGTs (0.88 ± 0.03) dominate over their OH-GDGT counterparts (0.12 ± 0.03) (Fig. 6b). GDGTs and OH-GDGTs with no cyclopentane moieties are the most dominant (e.g., GDGT-0 and OH-GDGT-0), followed by crenarchaeol and then the GDGTs and OH-GDGTs with 1, 2, and 3 cyclopentane moieties (Fig. 6c-d). Spatially, the fractional abundance of GDGTs tends to be relatively lower within the modern limits of the NOW, whereas the fractional abundance of OH-GDGTs is relatively higher within the modern limits of the NOW (Fig. S11).

In terms of regressions against different environmental variables (e.g., temperature, salinity, dissolved oxygen, and nitrate), we find that the tested GDGT- (TEX$_{86}$ and TEX$_{86}^L$) and OH-GDGT-based indices (RI-OH and RI-OH') range in the strength of the correlation coefficients ($R^2$) and significance ($p$ values) (Figs. 7, S12-S14). Moreover, differences in the season and depth integration of the environmental variables also appear to influence the strength of their correlation (Figs. 7, S12-14). For TEX$_{86}$, the strongest significant temperature correlation is achieved with summer SST in the upper 20 m of the water column ($R^2$ = 0.21 to 0.35, Fig. 7a). Although salinity and dissolved oxygen seem to have little confounding influence ($R^2 <$ 0.22), we find a moderately strong but significant correlation between annual TEX$_{86}$ and nitrate concentrations above 5 m bsl ($R^2$ = 0.40, Fig. S14). For TEX$_{86}^L$, the strongest significant temperature correlations ($R^2 > 0.40$) are found with annual temperature (>30 m bsl, Fig. 7b). The influence of other environmental variables (e.g., salinity, dissolved oxygen, and nitrate)

on $TEX_{86}^L$ appears to be generally weak ($R^2 < 0.27$), although summer nitrate concentration correlations at 0-200 m bsl are slightly higher and significant ($R^2 = 0.37$, Fig. S14). For RI-OH, the strongest significant temperature correlations are achieved with annual SST between 10 and 20 m bsl ($R^2 = 0.44$ to $0.46$). Autumn subsurface temperature between 60 and 80 m bsl also feature similarly significant correlations ($R^2 = 0.41$ to $0.42$) (Fig. 7c). The influence of the other tested environmental variables on RI-OH appears to be generally weak ($R^2 < 0.27$), although summer and annual 0-20 m dissolved oxygen concentration correlations are slightly higher ($R^2 = 0.39$ and $R^2 = 0.36$, respectively, Fig. S13). Finally, for RI-OH', the strongest significant temperature correlations are achieved with annual 0-20, annual 0-30 m SST, and autumn 0-100 m SST, although correlation coefficients for all are relatively low ($R^2 = 0.23$) (Fig. 7d). Based on our $R^2$ values, the influence of other environmental variables, namely annual and summer subsurface dissolved oxygen and nitrate below 100 m depth, appears to exert relatively more influence than temperature on RI-OH' (Figs. S13-14).

For the two indices that feature the strongest significant correlations with temperature ($R^2 > 0.40$), we derived individual linear temperature calibrations for Baffin Bay based on this study's dataset ($n=13$). The $TEX_{86}^L$ calibration ($R^2 = 0.45$) encompasses the surface and subsurface waters from 0-90 m bsl and features a standard error (S.E.) of 0.13 °C (Fig. 8a):

$$subT = 10.573x + 6.636, \tag{10}$$

The RI-OH calibration ($R^2 = 0.46$) encompasses the surface waters from 0-20 m bsl and features a S.E. of 0.26 °C (Fig. 8b):

$$SST = 27.982x - 31.524 \tag{11}$$

## 6. Discussion

### 6.1. Spatial variability in surface productivity

A recent and expanded Arctic study by Kolling et al. (2020) explored the efficacy of using a variety of sedimentary HBIs (i.e., $IP_{25}$, HBI II, and HBI III), sterols (i.e., brassicasterol and dinosterol) and their $PIP_{25}$-derived indices to reconstruct sea ice extent/concentration and pelagic productivity by comparison to satellite data. For Baffin Bay, the main conclusions were that $P_BIP_{25}$, $P_DIP_{25}$ and $P_{III}IP_{25}$ all exhibited strong correlations with modern spring and autumn seasonal sea ice concentration (Kolling et al., 2020). On the other hand, $TR_{25}$, the recently proposed HBI proxy for spring phytoplankton bloom in the Barents Sea (Belt et al., 2019), did not exhibit a clear relationship with chlorophyll $a$, but instead paralleled spring/autumn/winter sea ice extent (Kolling et al., 2020). For this reason, we did not further explore the controls of chlorophyll $a$, which is a limited and discontinuous dataset in the Baffin Bay region (NASA MODIS), on the distribution of HBI III and HBI IV compounds that comprise the $TR_{25}$ proxy (e.g., Belt et al., 2019). However, we do further explore the ability of HBIs and $PIP_{25}$ indices to track sea ice extent throughout Baffin Bay and include another previously untested $PIP_{25}$ index ($P_{IV}IP_{25}$). Most importantly, we explore the ability of HBI and sterol biomarkers, including two additional sterols (ß-sitosterol and campesterol) not analyzed by Kolling et al. (2020), to track the high surface productivity associated with the NOW. While it is common to report HBI and sterol concentrations relative to TOC, we note that only a subset of this study's dataset ($n=9/13$) and that of Kolling et al. (2020, $n=51/70$) have corresponding TOC measurements. Given that the overall pattern of HBI and sterol concentrations in the NOW compared to sites outside the NOW is the same whether concentrations are normalized to g

sediment or TOC in both datasets (Figs. 4a-d and S5), we focus the following discussion on samples normalized to g sediment to present a larger combined dataset ($n$=83). We note that while the methods to extract and separate biomarkers differ between Kolling et al. (2020) and this study, each method is routinely used and unlikely to result in any substantial dataset differences

(e.g., Belt et al., 2014).

In terms of HBIs, mean concentrations of all compounds are higher in sites within the NOW compared to sites outside the NOW in both datasets (Fig. 4a-b). However, only the concentration of HBI II is significantly different between NOW and non-NOW sites across the two datasets (Fig. 4a-b), while HBI IV is significantly different in this study (Fig. 4a) and HBI III is significantly different in Kolling et al. (2020) (Fig. 4b). For $IP_{25}$ and HBI II, the detection of each in all samples is consistent

with the spring and autumn sea ice limits in Baffin Bay (Fig. S6, Cavalieri et al., 1996; Bi et al., 2019). We also observe a strong positive correlation between $IP_{25}$ and HBI II (Fig. 5a-b), which likely supports a common sympagic diatom source as observed elsewhere in the Arctic (Belt et al., 2008; Cabedo-Sanz et al., 2013; Brown et al., 2014; Limoges et al., 2018). Given the distinct oceanographic seasonal characteristics of the NOW (open water) and central Baffin Bay (seasonal sea ice), we expected higher $IP_{25}$ and HBI II concentration in the sites south of the NOW compared to sites within the NOW itself – an

hypothesis inconsistent with our observations. One possibility is that these HBIs are produced earlier in the spring or later in the autumn when sea ice covers both the NOW and east coast of Baffin Island (Bi et al., 2019). Another possibility, may be that the higher concentrations of $IP_{25}$ and HBI II within the NOW reflect the southward transport of drift ice through the open NOW as well as the formation and export of thin sea ice from the NOW (Bi et al., 2019). While HBI III and IV are found in relatively lower concentrations than $IP_{25}$ and HBI II (Fig. 4a-b), all four HBI's distributions are highly correlated with one

another and spatially similar (Figs. 5 and S6). This presents several possibilities. First, as observed in other regions of the Arctic such as the Barents Sea, increased concentrations of HBI III and IV are associated with populations of open water (pelagic) phytoplankton in Marginal Ice Zones (Smik et al., 2016; Belt et al., 2015, 2019). In this sense, the concentrations we observe in Baffin Bay are consistent with increased production of these HBIs during the spring transition from seasonal sea ice to open water in the NOW compared to sites outside that maintain seasonal sea ice cover during the summer. Alternatively,

recent work that has monitored the production of various HBIs through the spring sea ice melt season in southwest Baffin Bay shows that HBI III is produced under the sea ice before and concurrent with $IP_{25}$, and therefore, is likely biosynthesized by sea ice (sympagic) diatoms in this region (Amiraux et al., 2019, 2021). Given that HBI III and IV in both datasets show weak or insignificant correlations with sterols (Fig. 5a-b), which are generally attributed to open water phytoplankton (e.g., Köseoğlu et al., 2019), we argue that the latter scenario where HBI III and IV are produced by sympagic diatoms is most likely.

In terms of sterols, mean concentrations of all compounds were higher and statistically different for sites in the NOW compared to sites outside the NOW in both datasets (Fig. 4c-d). We note that the non-NOW data for brassicasterol and dinosterol from Kolling et al. (2020) are proportionally higher than ours, which may reflect the incorporation of a greater diversity of oceanographic settings in Baffin Bay compared to ours. While a recent Holocene marine record from Petermann Fjord (Northwest Greenland) interprets campesterol and ß-sitosterol as indicators of terrestrial input (Detlef et al., 2021), we

note that all sterols are highly correlated in our dataset (Fig. 5a), which suggests a possible common source. Considering our

bulk geochemistry data indicates a predominance of marine organic matter origin (Fig. 3a), the known production of all four sterols by marine diatoms (Belt et al., 2013, 2018; Rampen et al., 2010), and the low terrestrial biomass of surrounding landmasses (Gould et al., 2003), we suggest that in Baffin Bay, a marine source is most likely for brassicasterol, dinosterol, campesterol and ß-sitosterol. Although these sterols do not feature the same degree of source specificity as the HBIs do, the statistically higher concentrations of sterols in the NOW in both datasets is consistent with the high seasonal pelagic productivity that the NOW supports today (Fig. 4c-d). Moreover, studies based on nutrient concentrations of particulate matter in Baffin Bay show that primary productivity in the NOW is an order of magnitude higher than adjacent areas in Baffin Bay during spring/summer (Tremblay et al., 2002a). Given the southward propagation of sea ice, and that the development of the NOW occurs at the transition between spring and summer (Bi et al., 2019), the sterols are likely recording a spring/summer season signal of pelagic productivity.

Finally, in terms of $PIP_{25}$, all four indices ($P_{III}IP_{25}$, $P_{IV}IP_{25}$, $P_BIP_{25}$, and $P_DIP_{25}$) for our samples were lower in sites within the NOW compared to sites outside the NOW (Fig. 4e). $P_BIP_{25}$ and $P_DIP_{25}$ values from Kolling et al. (2020) are higher in the NOW compared to non-NOW sites, while $P_{III}IP_{25}$ and $P_{IV}IP_{25}$ are consistent with ours (Fig. 4f). However, only the $P_DIP_{25}$ indices in both datasets are statistically different for sites in the NOW (Fig. 4e-f). The difference in $P_BIP_{25}$ and $P_DIP_{25}$ between datasets likely stems from the larger diversity of sites included in Kolling et al. (2020, $n$=70) compared to our smaller dataset ($n$=13). While the HBI-derived indices ($P_{III}IP_{25}$ and $P_{IV}IP_{25}$) feature overall higher values compared to those derived from sterols ($P_BIP_{25}$, and $P_DIP_{25}$) in both datasets, we caution against the application of the HBI-derived indices in Baffin Bay for sea ice concentration since these biomarkers may partly originate from sea ice diatoms (e.g., Amiraux et al., 2019, 2021), rather than a combination of sea ice and open water diatoms. Moreover, these indices show no statistical difference between NOW and non-NOW sites (Fig. 4e-f). Looking at the only statistically different sterol-derived index (i.e., $P_DIP_{25}$) for both datasets, the trends are opposite, where this study shows higher concentrations of sea ice outside the NOW and Kolling et al. (2020) show higher concentrations of sea ice inside the NOW (Fig. 4e-f). This difference stems from the datasets' dinosterol concentrations, which show less variability between NOW and non-NOW sites in our dataset compared to Kolling et al. (2020) (Fig. 4c-d), likely due to the inherently larger spatial coverage of the latter dataset. While it seems logical to therefore rely on the distribution of $P_DIP_{25}$ values from Kolling et al. (2020), the spatial variability of the dataset is inconsistent with late spring and summer sea ice cover in Baffin Bay, where the highest concentrations of sea ice are found off eastern and northeastern Baffin Island (Bi et al., 2019). Given the limited size of our dataset, which is consistent with these observations, we acknowledge that more work is needed to assess the utility of PIP indices in Baffin Bay.

In summary, we have several recommendations for future lipid-based paleoenvironmental reconstructions in Arctic oceanographic settings. First, for complex oceanographic settings like Baffin Bay, we recommend the analysis of both sterols and HBIs to test the performance of various sea ice cover proxies and their ability to track local changes in sea ice and pelagic productivity. If possible, this would be best achieved through the analysis of modern surface sediments and sediment traps over an environmental gradient and through the seasons to capture proxy response to known variable changes (e.g., Navarro et al., 2013; Smik and Belt, 2017; Koch et al., 2020). Second, continued research on sterol and HBI sources and seasonality of

production is critical for the development of more refined sea ice and marine productivity reconstructions (e.g., Limoges et al., 2018; Amiraux et al., 2019, 2021). This is particularly necessary when combining $IP_{25}$, a known sea ice diatom proxy, with less source specific open water proxies such as HBI III and sterols. Third, based on replicated Baffin Bay datasets, which show statistically higher sterol concentrations within the NOW compared to sites outside of its modern limits (this study, Kolling et al., 2020), we suggest that sterols are a key tool to characterize the presence/absence of the NOW in the recent geologic past.

The complimentary analysis of HBIs can then provide important context for sea ice changes, and how that may have impacted pelagic productivity. Finally, the integration of additional paleoenvironmental proxy datasets, such as microfossils (e.g., foraminifera, diatoms, dinocysts) and bulk geochemistry, can lead to even more detailed reconstructions of past pelagic productivity and how that relates to various environmental drivers (e.g., Jackson et al., 2021; Ribeiro et al., 2021).

## 6.2. Temperature calibrations

Temperature correlations for GDGTs first relied on empirical correlations between global surface sediments and the variability in biomarker structure (e.g., Schouten et al., 2002). Subsequent iterations and developments of $TEX_{86}$-based indices removed some compounds (e.g., crenarchaeol stereoisomer for $TEX_{86}^L$) for improved performance in certain latitudinal bands (Kim et al., 2012). Most recently, Bayesian statistics have been employed to generate spatially varying calibrations (i.e., BAYSPAR)

that reach a compromise between data-constrained global and regional calibrations to capture regional oceanographic variability and site-specific uncertainty more accurately (Tierney and Tingley, 2014). However, BAYSPAR does not include surface sediment for sensitive oceanographic regions like Baffin Bay (Tierney and Tingley, 2014). For this reason, developing local correlation-constrained calibrations can be particularly useful to capture the nuances of oceanographic variability in regions where global dataset coverage is lacking or where the temperature relationship deviates from global linear calibrations

(e.g., Harning et al., 2019; Park et al., 2019; Fietz et al., 2020; Sinninghe Damsté et al., 2022). Moreover, exploring the potential influence of additional environmental variables (e.g., salinity, DO, nutrients) will help us better understand the mechanisms behind and functionality of this commonly applied paleotemperature proxy. Given the relatively small sample size ($n$=13) and narrow temperature range of our dataset (~6 °C, Fig. 2a) compared to global compilations (e.g., Kim et al., 2012; Lü et al., 2015), we acknowledge that our conclusions in the following discussion are relatively limited. However, as no previous GDGT

and OH-GDGT exists from this region, it is an important first step in describing their distribution and relationships with environmental variables.

### 6.2.1. GDGTs and OH-GDGTs

While a more detailed analysis of intact polar lipid production and genetic diversity in Baffin Bay is lacking, the distribution

of GDGT and OH-GDGT in our study area (Fig. 6) and understanding of their production in cultures (Elling et al., 2017) indicate that planktic group 1.1a Thaumarchaeota are likely the dominant producers. Therefore, the global relationship between $TEX_{86}$ and the RI (Fig. 6a, black polynomial line) serves as a simple means to evaluate whether the $TEX_{86}$-based indices are influenced by additional nonthermal factors (Zhang et al., 2016). Even though our data exhibit a correlation between $TEX_{86}$

and RI values ($R^2 = 0.46$) with a slope like the global polynomial regression, all samples plot below the lower 95% uncertainty
limit (Fig. 6a). A recent study from the South China Sea found a similar relationship to ours, in that $TEX_{86}$ values were well
correlated with RI values, but did not conform to the global polynomial trend's uncertainty (Wei et al., 2020). Wei et al. (2020)
posited that the shallow shelf environment (neritic zone) of the South China Sea may result in the observed deviation from the
global polynomial $TEX_{86}$-RI relationship, as shallow water Thaumarchaeota respond differently to temperature than the deep-
dwelling communities (e.g., Kim et al., 2015, 2016; Villanueva et al., 2015; Zhu et al., 2016; Jia et al., 2017). While the depths
of our sites are deeper than the neritic zone (>200 m bsl, Table 1), many are much shallower than open ocean sites used in the
global calibrations (Kim et al., 2010, 2012). One possibility is that the shallower environment of northern Baffin Bay may
result in a different response of Thaumarchaeota to temperature and different distribution of GDGTs than in the open ocean.
In any case, temperature seems to remain the dominant control on GDGT cyclization in this region.

Our regression analysis against temperature, salinity, DO and nitrate further supports temperature as the dominant
environmental control on GDGT distributions in Baffin Bay for the seasons available in the WOA18 dataset. The lack of
WOA18 winter temperatures, in addition to the fragmentary dataset for spring temperatures in Baffin Bay (Locarnini et al.,
2018), prevents us from assessing the impact of these individual seasons, which is unfortunate given that cold season
temperatures in other high-latitude settings exhibit a stronger correlation with GDGT distributions (Herfort et al., 2006; Rueda
et al., 2009; Rodrigo-Gámiz et al., 2015; Harning et al., 2019). However, the higher correlation between $TEX_{86}^L$ and annual
temperatures compared to summer/autumn seasons (Fig. 7b) suggests a bias for Baffin Bay GDGT production towards the
cooler seasons. Although we explored the relationship of the original $TEX_{86}$ index, we prefer to rely on the low-temperature
$TEX_{86}^L$ modification to assess GDGT depth and season and temperature relationships. This decision is supported by 1)
generally low correlation coefficients for temperature and $TEX_{86}$, except for summer SST (Fig. 7a), and 2) the better
correlations of $TEX_{86}^L$ and lower water depths where ammonia-oxidation likely occurs (Fig. 6b, e.g., Hurley et al., 2018, Park
et al., 2019). The moderate correlation between $TEX_{86}$ and summer SST, and the lack of correlation between $TEX_{86}^L$ and SST
from any season, may indicate that the crenarchaeol stereoisomer, which is included in the $TEX_{86}$ index but not in $TEX_{86}^L$,
may be physiologically advantageous for surface dwelling Thaumarchaeota during the warmer summer months in Baffin Bay.
At least for one Thaumarchaeotal culture (*Candidatus* Nitrosotenuis uzonensis), temperature exerts a strong control on the
proportion of the crenarchaeol stereoisomer (Bale et al., 2019). However, we also cannot rule out the possibility of a yet
unknown alternative biological source, as phylogeny also exerts a secondary control (e.g., Bale et al., 2019).

In terms of other tested environmental variables, correlations between $TEX_{86}^L$ and salinity and dissolved oxygen are
generally poor ($R^2 < 0.3$, Figs. S12-13), consistent with previous reports on the lack of relationship between GDGT production
and salinity (Wuchter et al., 2004, 2005; Elling et al., 2015). The presence of a well-oxygenated water column in Baffin Bay
(Fig. 2c) likely has little influence on GDGT cyclization, which has been reported to only occur in oxygen-limited
environments (e.g., Qin et al., 2015). The correlation between $TEX_{86}^L$ and nitrate, while weak, is more pronounced in the
lowermost integrated depth (200 m bsl) (Fig. S14). The main product of ammonia-oxidation is nitrite, for which we have no
data for in this region. However, since nitrite can be subsequently oxidized to nitrate by bacteria (Kuypers et al., 2018), we

assume that nitrate provides indirect evidence for both reactions. This provides evidence for a subsurface depth habitat of ammonia oxidizing Thaumarchaeota in Baffin Bay, which supports our observations that $TEX_{86}^L$ correlates best with annual subT (Fig. 8c).


Our complimentary analysis of OH-GDGTs and environmental variables reveals several differences with the conclusions drawn for GDGTs in Baffin Bay and with OH-GDGTs elsewhere. First, the RI-OH' index, which was developed as a modified version of RI-OH for low-temperature environments (Lü et al., 2015), does not perform as well as the RI-OH index in our dataset (Fig. 7c-d). Second, we find that RI-OH is best correlated with annual SST in the top 40 m of the water column ($R^2 = 0.46$), and with autumn in the subsurface (>50 m depth, Fig. 7c). The slightly stronger correlation between RI-OH and SST compared to subT is supported by Lü et al. (2019), who observed higher concentrations of OH-GDGTs in the upper portion of the water column compared to GDGTs in the East China Sea. Moreover, the slightly higher proportion of OH-GDGTs relative to GDGTs in our samples within the modern limits of the NOW (Fig. S11), where sea ice concentrations are low and sea surface primary productivity is high, suggests that these surface conditions may be conducive to their production in Baffin Bay. Additional studies have found that the ring composition of OH-GDGT often differs from GDGTs, which may suggest that these two lipid classes are sourced from different Thaumarchaeota subgroups or produced in different niches of the water column characterized by different environmental conditions (Liu et al., 2012).



In terms of additional environmental variables, only surface dissolved oxygen appears to exert a partial influence on OH-GDGT cyclization in the RI-OH index (Fig. S13c). However, when OH-GDGT-0 is considered in the RI-OH' index, moderate correlations with dissolved oxygen (Fig. S13d) and nitrate appear within the subsurface waters (Fig. S14d). These latter observations conflict with our RI-OH temperature correlations that suggest a surface-dwelling producer of OH-GDGTs. Instead, this may suggest that OH-GDGT-0 is important for membrane functionality at these depths and/or for these environmental variables. Alternatively, OH-GDGT-0 may also have a different biological source in Baffin Bay, as at least for GDGTs, GDGT-0 can be produced by other groups of archaea (e.g., Pearson and Ingalls, 2013). Given the relatively low correlations between RI-OH', which includes OH-GDGT-0, and temperature, our data further supports the potential role of OH-GDGT-0 for membrane response to these non-thermal environmental variables in Baffin Bay. Although our data suggests that OH-GDGTs, reflected by the RI-OH index, are predominately a proxy for annual SST, more detailed studies on the biological producers, depth habitat, and response to environmental variables of OH-GDGTs will undoubtedly benefit future applications and interpretations of downcore OH-GDGT proxy records.



Following our evaluation of GDGT and OH-GDGT indices in terms of their ability to capture temperature, amongst other environmental variables, we present two temperature calibrations that can benefit future paleoceanographic reconstructions from Baffin Bay (Fig. 8). Like other regional $TEX_{86}^L$ temperature calibrations in the greater North Atlantic region (e.g., Harning et al., 2019), our Baffin Bay $TEX_{86}^L$ temperature calibration captures a subsurface signal (0-90 m bsl) with a considerably lower S.E. (0.13 ℃) compared to the latest global calibration (4.0 ℃, Kim et al., 2012) despite the lower $R^2$ (0.45 vs. 0.87, respectively). While other subsurface depth calibrations are nearly as strong for 0-90 m bsl, which simply reflects the isothermal nature of temperature below ~80 m at our study sites (Fig. 2a), we choose to focus on the strongest for



the sake of simplicity. The strength of our local $TEX_{86}^L$ subT calibration is also supported by the fact that other regional (Harning et al., 2019) and global calibrations (Kim et al., 2012) overestimate the observed WOA18 temperature in Baffin Bay (Fig. 8c), with residuals as high as 5.8 °C for the Iceland calibration and 5.0 °C for the global calibration (Fig. 8e). In terms of
the RI-OH SST calibration, to the best of our knowledge there are no regional calibrations to compare with. However, comparison with the global calibration that features a S.E. of 6.0 °C (Lü et al., 2015) again reveals the power of reducing uncertainty with regional calibrations as ours from Baffin Bay features a smaller S.E. of 0.26 °C. While the global $TEX_{86}^L$ calibration overestimates Baffin Bay temperatures by up to 5.0 °C, the RI-OH calibration overestimates local temperatures by less than 3.3 °C (Fig. 8d-f). In future studies, we suggest that these two calibrations may help quantify paleotemperature
changes in Baffin Bay surface and subsurface waters. For other high-latitude locations that do not yet have local GDGT-temperature calibrations, our comparison between instrumental temperature and global calibration estimates caution against quantitative interpretations due to large over estimations. However, as the trends between regional and global calibrations are similar (Fig. 8c-d), qualitative interpretations remain valid.

## 7. Conclusions

As global climate change continues, the sustainability of Arctic polynyas is in jeopardy. While proxy reconstructions of polynyas prior to the instrumental period can shed light on key climate and environmental mechanisms that lead to polynya presence/absence, continued development of our understanding of those proxies is needed. Therefore, we evaluated a series of lipid biomarkers (HBIs, sterols, GDGTs, and OH-GDGTs) in surface sediment samples from Baffin Bay to characterize how these biomarkers capture sea ice and productivity conditions in the North Water Polynya (NOW) as well as inform the utility
of commonly applied paleotemperature proxies in Baffin Bay. In conjunction with a similar study of lipid biomarker productivity proxies from the same region (Kolling et al., 2020), we draw the following conclusions:

- All studied HBIs ($IP_{25}$, HBI II, HBI III, and HBI IV) exhibit strong correlations with each other and are found in relatively higher concentrations within the modern limits of the NOW. Based on comparison with seasonal sea ice limits, we suggest all HBIs are, at least partially, produced by sympagic diatoms under the sea ice during spring and
autumn.
- All studied sterols (brassicasterol, dinosterol, campesterol, and ß-sitosterol) exhibit statistically higher concentrations within the NOW compared to sites further south in Baffin Bay, consistent with the order of magnitude higher spring/summer primary productivity that is observed within the NOW today relative to surrounding waters. Hence, we suggest that sterols are well-suited for open water paleoproductivity reconstructions in this region, with particular
emphasis on surface productivity related to the NOW.
- Application of the GDGT-based $TEX_{86}^L$ index is optimal over $TEX_{86}$ in Baffin Bay and captures an integration of annual subT consistent with the depth habitat of Thaumarchaeota observed elsewhere around the globe. Other tested environmental variables, such as salinity, dissolved oxygen, and nitrate reveal low correlations, which we infer to

reflect negligible influence on GDGT cyclization. We present a local annual subT calibration that provides a lower S.E. (0.13 °C) compared to the latest global calibration (4.0 °C, Kim et al., 2012).

- Application of the OH-GDGT-based RI-OH index is optimal over RI-OH', even though the latter was originally developed for low temperature environments. The RI-OH index captures an integration of annual SST, which may suggest that OH-GDGTs in Baffin Bay are biosynthesized by different microbes than those that produce GDGTs and/or are sensitive to different environmental stressors. Our local annual SST calibration provides a lower S.E. (0.26 °C) compared to the latest global calibration (6.0 °C, Lü et al., 2015).

- Future multi-proxy approaches that include, but are not limited to, these lipid biomarker toolsets in marine sediment core studies from the NOW will allow for more refined and detailed interpretations on the polynya's past inception and stability throughout Earth's history.

## Data Availability

After acceptance, all data will be stored on the PANGAEA repository.

## Author Contributions

JS and DJH designed the study and JS and AEJ funded the study. DJH led the analyses of samples and developed GC-MS methods under the supervision of JS. BH and LW assisted with the extraction and purification of samples. DJH wrote the manuscript with discussion and contribution from all co-authors.

## Competing Interests

The authors declare they have no conflicts of interest.

## Acknowledgements

The authors are grateful to the Inuit of Inuit Nunangat (Canada), the Kalaalit of Kalaalit Nunaat (Greenland), and the Inughuit of Avanersuarmiut (West Greenland) for access to their homelands to conduct this research. We kindly thank the captains, crews, and scientific staffs aboard the 2008 CSS *Hudson* and 2015 and 2017 CSS *Amundsen* research cruises for their efforts in collecting the surface sediment samples and Drs Simon Belt and Lukas Smik for providing 7-HND and 9-OHD standards. We appreciate the valuable analytical support of Dr Nadia Dildar at the University of Colorado Boulder and suggestions from Dr Ruediger Stein and an anonymous reviewer on an earlier version of this manuscript.

## Financial support

This project has been supported by the National Science Foundation grant ARN-1804504.

**Review Statement**

This paper was edited by Helge Niemann and reviewed by Darci Rush, Sofia Ribeiro and one anonymous referee.

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

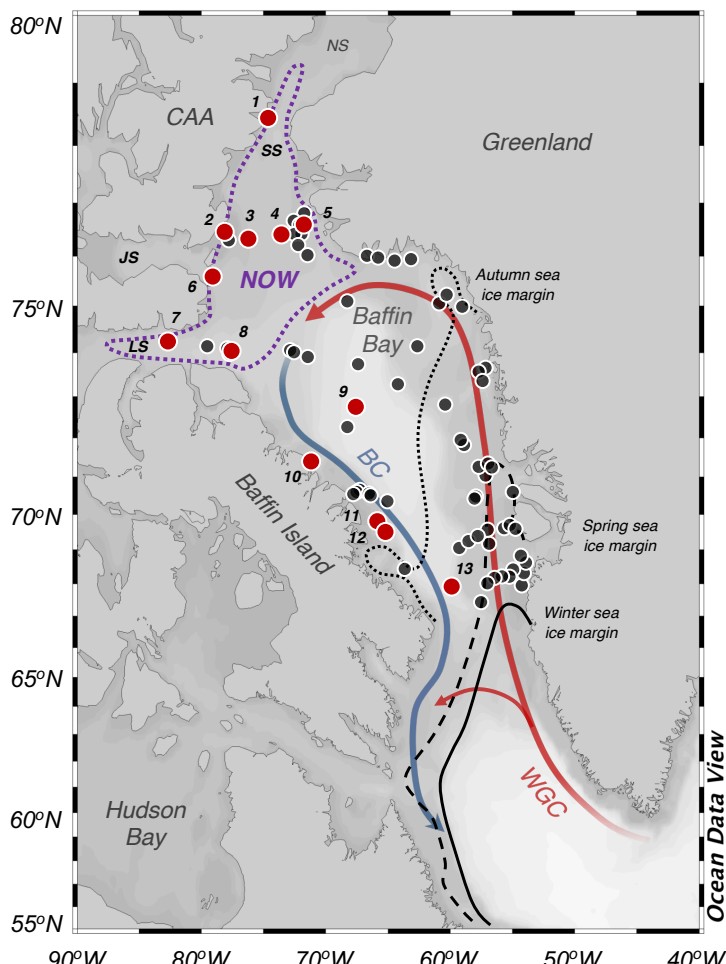

**Figure 1: Overview map of Baffin Bay. Simplified ocean surface currents shown in bold lines where red reflects warm, Atlantic Water (West Greenland Current, WGC) and blue reflects cool, Arctic Water (Baffin Current, BC). To the north is the June limit of the NOW (purple dotted line). Seasonal sea ice limits shown with black dotted (autumn), dashed (spring) and solid lines (winter) (Cavalieri et al., 1996). The numbered locations of this study's modern sites are shown with red circles (*n*=13), and those of Kolling et al. (2020, *n*=70) in black. CAA = Canadian Arctic Archipelago, NS = Nares Strait, SS = Smith Sound, JS = Jones Sound, and LS = Lancaster Sound. See Table 1 for further sample site information. Ocean Data View base map after Schlitzer (2019).**

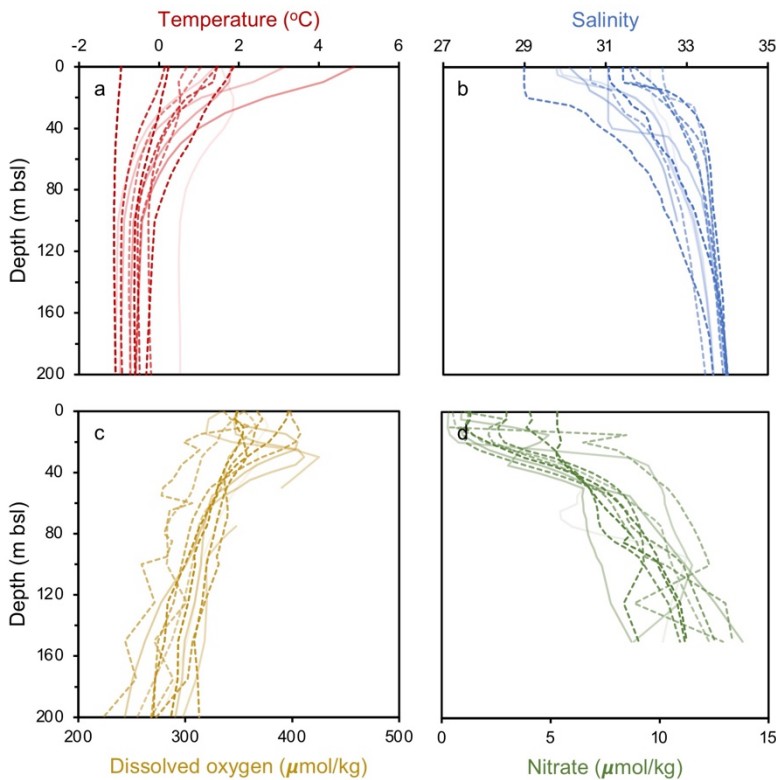

**Figure 2: WOA18 Annual 2007-2017 oceanographic variables from Baffin Bay against depth (m bsl). Individual profiles are from each study site, where darker (lighter) colors reflect sites farther north (south) and dashed (solid) lines denote those within (outside) the modern limits of the NOW. Data from Garcia et al. (2018a, 2018b), Locarnini et al. (2018), and Zweng et al. (2018).**


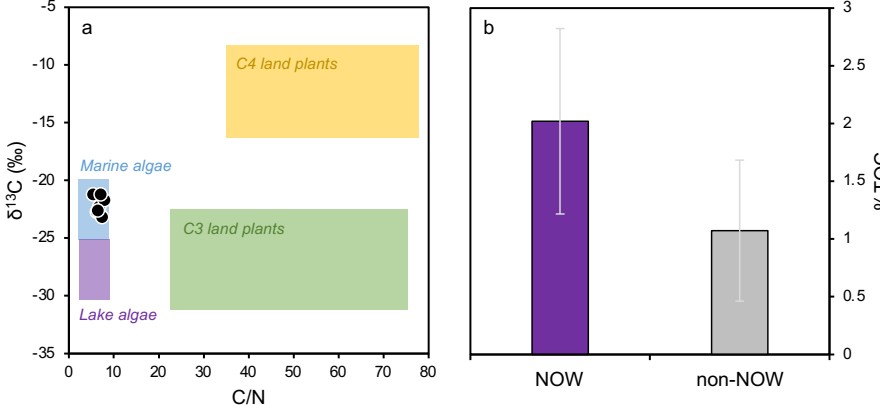

**Figure 3: Bulk geochemistry data for this study's dataset (*n*=9, Table 1). a) organic matter provenance based on bulk C/N and δ¹³C values, where reference values for four endmembers are after Meyers (1994), and b) average %TOC and standard deviation for sites in the NOW (purple, *n*=5) and sites outside the NOW (gray, *n*=4).**


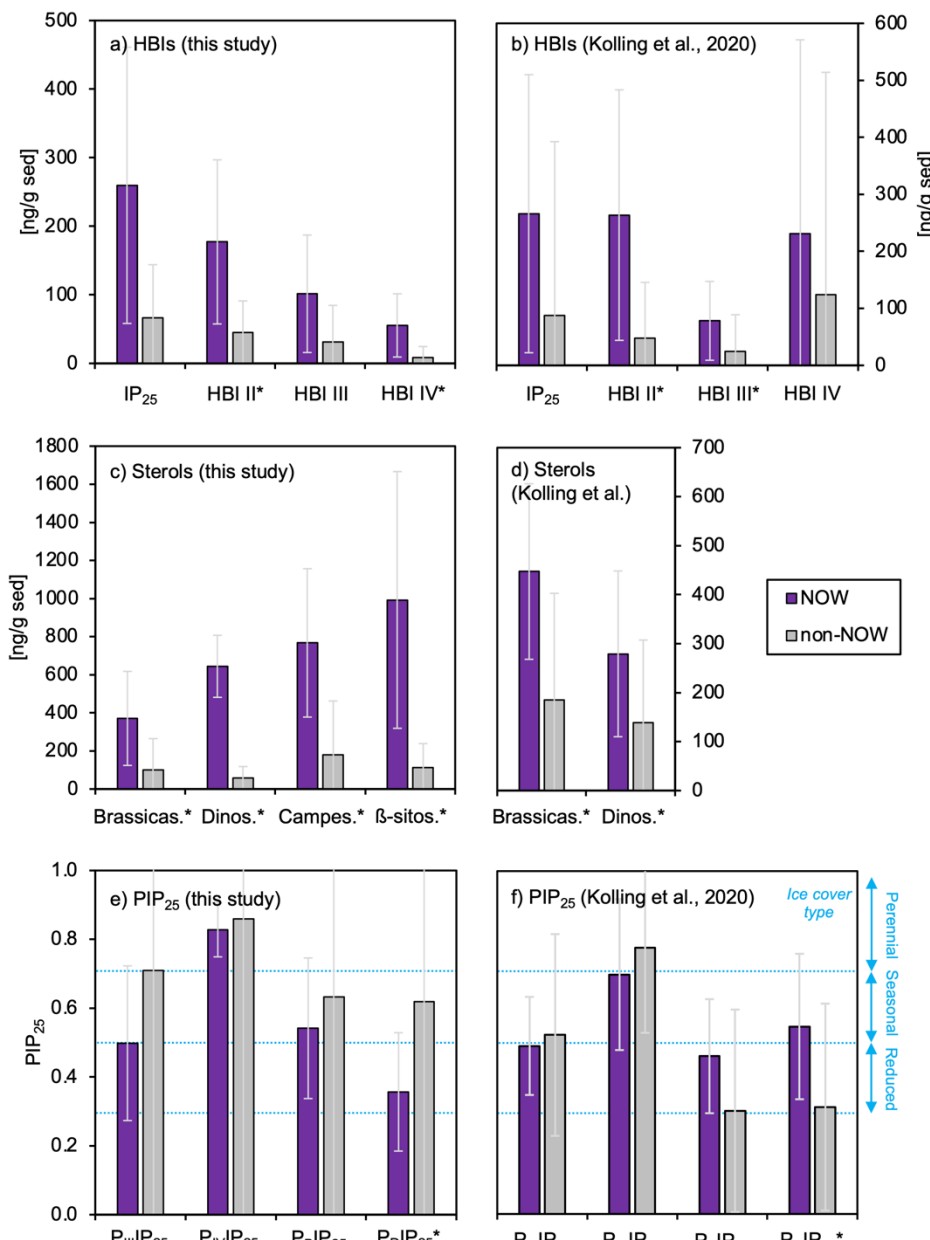

**Figure 4: Average concentrations (ng/g sed) and standard deviation of a-b) HBIs, c-d) sterols, and e-f) PIP$_{25}$ indices for sample sets from this study (*n*=13) and the subset of Kolling et al. (2020, *n*=70). Standard deviations for each shown in light gray. Sites within the NOW are colored purple whereas sites outside the NOW are gray. Qualitative sea ice concentration limits in panel e-f after Müller et al. (2011) for no sea ice (0 to 0.3), reduced sea ice (0.3 to 0.5), seasonal sea ice (0.5 to 0.7), and perennial sea ice (0.7 to 1). For all NOW and non-NOW comparisons, differences that are statistically different as defined by t-tests (*p*-value < 0.05) are denoted with an \*. See Figure S4 for complimentary plots of biomarker concentrations normalized to TOC.**


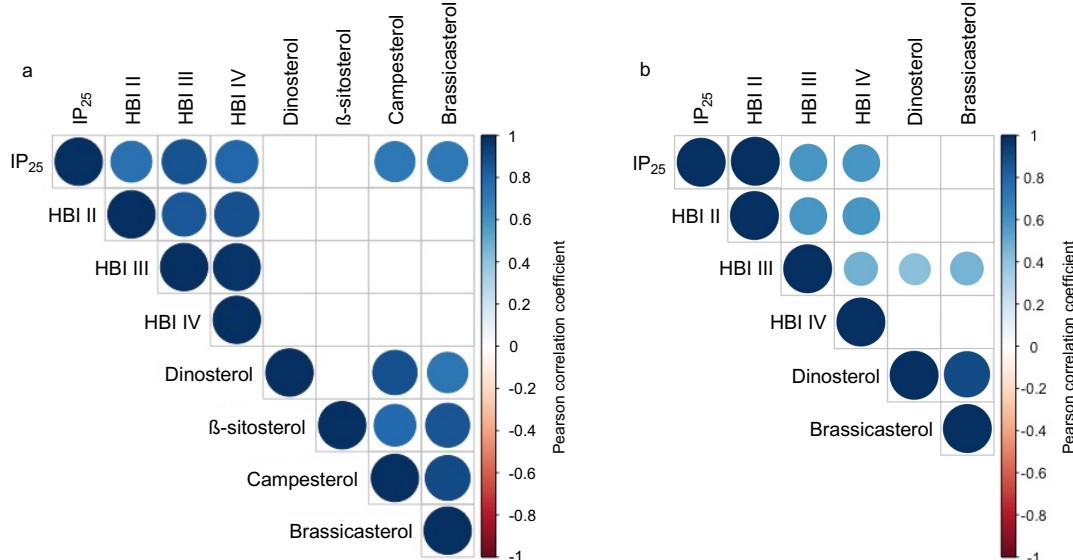


**Figure 5: Pearson correlation coefficients between HBIs and sterols for a) this study's samples (*n*=13) and b) the subset of Kolling et al. (2020, *n*=70). Positive correlations are displayed in blue and negative correlations in red. Both color and the size of the circle are proportional to the correlation coefficients. Insignificant correlations (*p*-values >0.01) are left blank.**

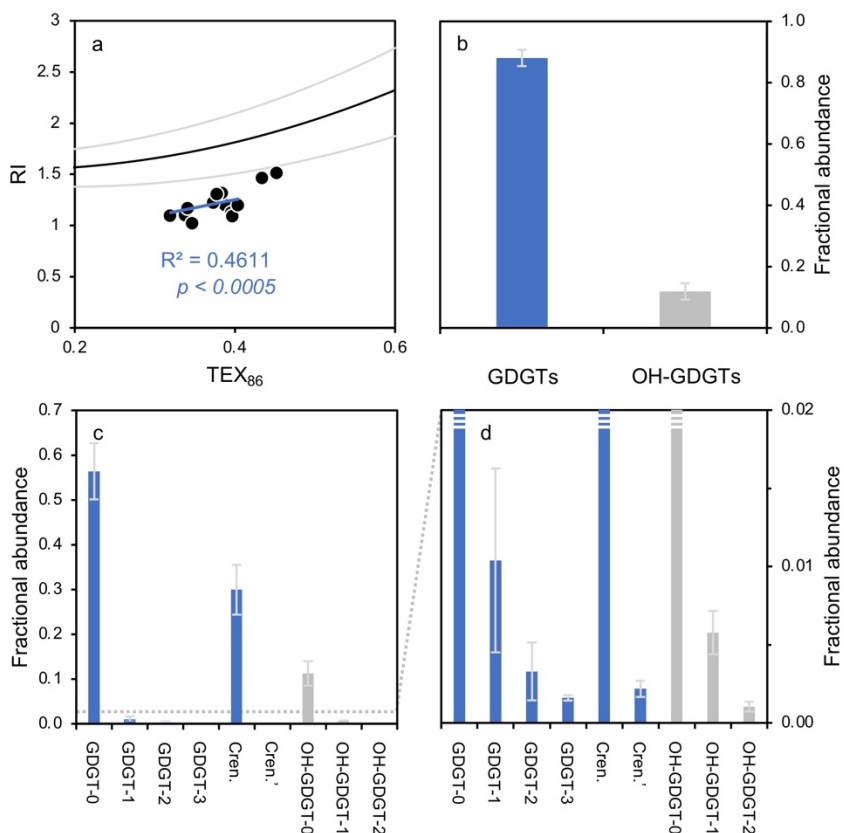


**Figure 6: GDGT- and OH-GDGT distributions and average fractional abundances and standard deviations for this study's dataset (*n*=13). a) TEX$_{86}$ versus RI showing the global polynomial equation and 95% uncertainty envelope (black and gray lines, Zhang et al., 2016) and Baffin Bay sediments (black dots), b) fractional abundance of total GDGTs (blue) and OH-GDGTs (gray), c) and d) fractional abundance of individual GDGTs (blue) and OH-GDGTs (gray) at two different scales.**


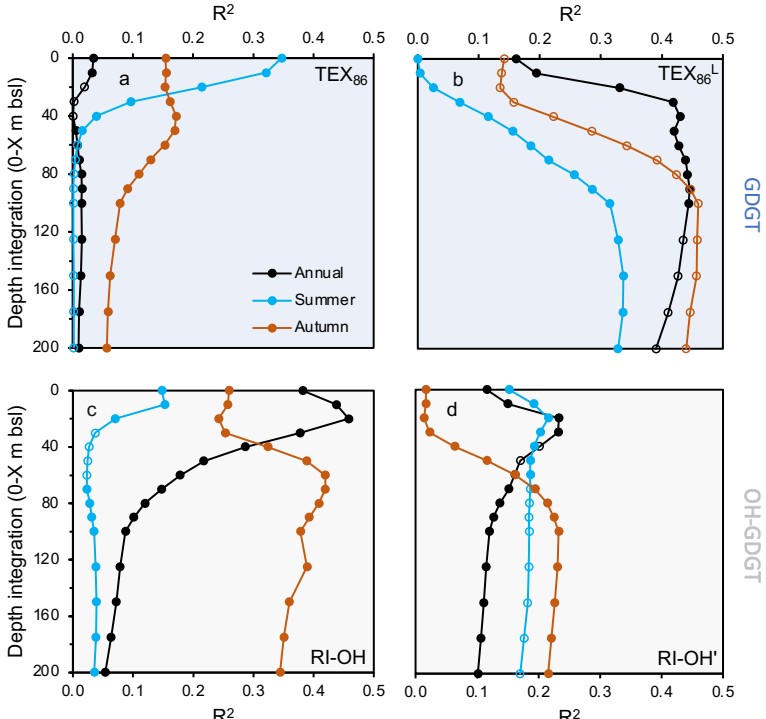

**Figure 7: Regression coefficients of GDGT-based temperature indices against WOA18 temperature at various depth integrations and seasons for this study's dataset (_n_=13). a) TEX₈₆, b) TEX₈₆ᴸ, c) RI-OH, and d) RI-OH'. WOA18 data from Locarnini et al. (2018). Data points not filled indicate insignificant correlations (_p_-value > 0.05).**

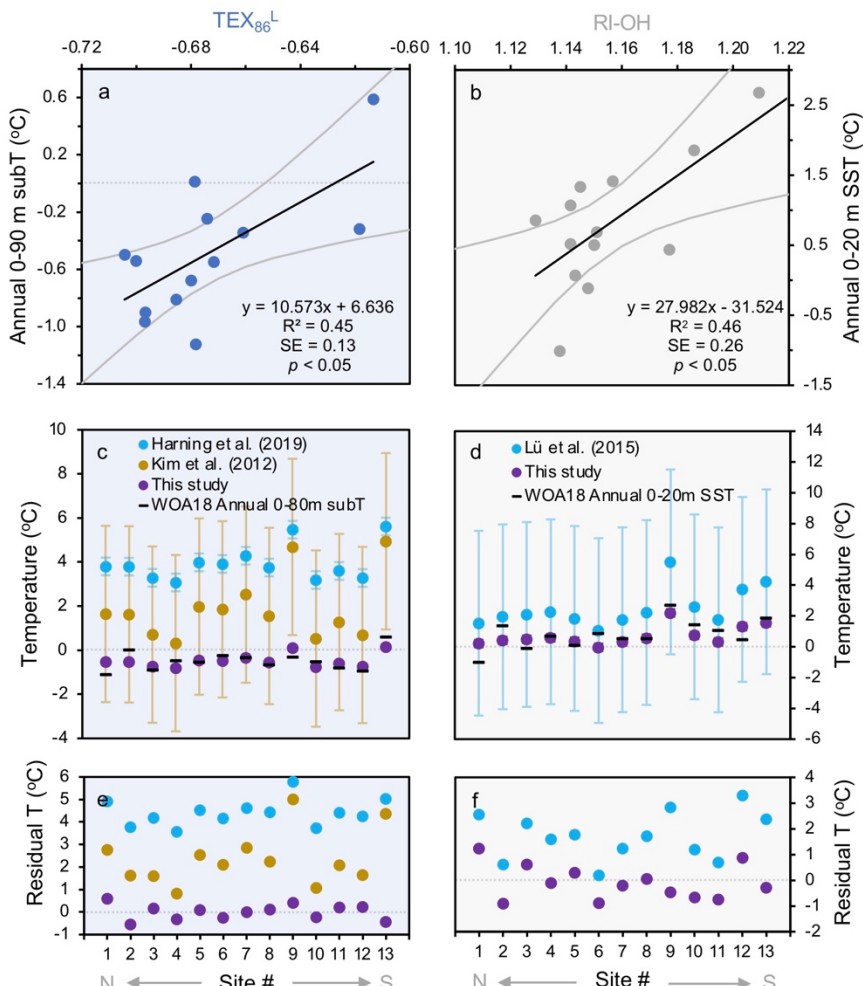

**Figure 8:** GDGT- and OH-GDGT-temperature calibrations for this study's dataset (*n*=13). Shown are a) TEX$_{86}^L$ vs annual 0-90 m subT, b) RI-OH (OH-GDGTs) vs annual 0-20 m SST, c) TEX$_{86}^L$ subT estimates based on calibrations from Iceland (blue, Harning et al., 2019), global (yellow, Kim et al., 2012), and Baffin Bay (purple, this study) against WOA18 annual 0-90m subT (black dashes, Locarnini et al., 2018), d) RI-OH SST estimates based on calibrations from global (blue, Lü et al., 2015) and Baffin Bay (purple, this study) against WOA18 annual 0-20m SST (black dashes, Locarnini et al., 2018). Panels e and f show residuals of calibration estimates and WOA18 data for TEX$_{86}^L$ and RI-OH, respectively. The x-axis refers to site # as shown in Fig. 1 and Table 1.

**Table 1: Marine surface sediment site information for this study.**

| Site # | Core Site | Lat | Long | Water depth (m bsl) | Collection year | Collection device | Analyzed for bulk geochemistry | Analyzed for lipid biomarkers |
|---|---|---|---|---|---|---|---|---|
| 1 | AMD17-129-BC | 78.42 | -74.24 | 521 | 2017 | Box core | X | X |
| 2 | AMD17-101-BC | 76.48 | -77.77 | 378 | 2017 | Box core | X | X |
| 3 | AMD17-108-BC | 76.47 | -74.70 | 449 | 2017 | Box core | X | X |
| 4 | AMD17-111-BC | 76.44 | -73.32 | 593 | 2017 | Box core | X | X |
| 5 | AMD17-115-BC | 76.57 | -71.33 | 668 | 2017 | Box core | X | X |
| 6 | HU2008 029-040 BX | 75.58 | -78.63 | 580 | 2008 | Box core | | X |
| 7 | HU2008 029-59 TC | 74.26 | -82.23 | 791 | 2008 | Trigger core | | X |
| 8 | HU2008 029-49 BX | 74.03 | -77.13 | 868 | 2008 | Box core | | X |
| 9 | AMD17-BB2-BC | 72.77 | -67.25 | 2373 | 2017 | Box core | X | X |
| 10 | AMD15-CASQ1-BC4 | 71.41 | -70.89 | 702 | 2015 | Box core | X | X |
| 11 | AMD17-176-BC | 69.82 | -65.46 | 267 | 2017 | Box core | | X |
| 12 | AMD17-Site 8.1-BC | 69.52 | -64.97 | 1054 | 2017 | Box core | X | X |
| 13 | AMD17-Disko Fan-BC | 67.99 | -59.60 | 1012 | 2017 | Box core | X | X |