# Peer review of "Biomarker characterization of the North Water Polynya, Baffin Bay: Implications for local sea ice and temperature proxies"

_Biogeosciences, 2022_

## Author Comment (AC1)

***Reviewer 1 (Darci Rush):***

This paper entitled "Biomarker characterization of the North Water Polynya, Baffin Bay: Implications for local sea ice and temperature proxies" submitted by Harning et al. presents the distribution of lipid biomarkers in marine surface sediments taken in the area in and surrounding the largest arctic polynya (North Water Polynya). They assess some of the most commonly evaluated lipid classes in organic geochemistry: highly branched isoprenoids, sterols, and GDGTs. Combining their results with data generated from previous studies in the area, these authors evaluate the use of highly branched isoprenoids and sterols as biomarkers for sea ice productivity. The authors also compare the estimated temperatures generated by biomarker (GDGT) proxies with measured data from the World Ocean Atlas Data from 2007 – 2017. The authors make several astute observations about the environmental controls that may govern the distribution of GDGTs involved in the temperature proxies in the Arctic. They have a reasonable argument for the power of regional calibrations in reducing unceratintity and errors in temperature estimates, but I feel that they must also acknowledge that the sample size of these observations is small. Overall, I am satisfied with their discussion of GDGT-based temperature proxies in the Arctic and sea ice cover/productivity and recommend this manuscript for publication. I do have a few comments for minor revisions, nevertheless:

*We greatly appreciate Darci Rush's time and consideration of our manuscript and thank the reviewer for a constructive critique that will lead to a stronger paper. Below we address each comment individually.*

**Results**

Line 257 onwards. It is not clear to me which sediment set you are discussing in terms of the GDGT and OH-GDGT data. Is it the 13 "Baffin Bay" samples collected for your study or are you including the 70 previously published samples? It would be good if you make the sample number clear in the text and in the figures. It is also confusing when the authors switch on line 283 to talking about "northern Baffin Bay". I assume that you mean just the 8 NOW sediments, but this needs clarity. Including number of samples with the correlation coefficients would make it clearer for the reader when you are examining 13 sediments, or just the 8 in the NOW sample subset and if you are also ever including previously published data. Indeed, if all your GDGT and OH-GDGT data analysis is for the 13 samples then you aren't saying anything with the GDGTs about the NOW (as you did do for the HBIs and sterols) but rather about the entire Baffin Bay area. This seems to deviate from the title and aim of the manuscript.

*Thank you for providing the opportunity to clarify this. The only available GDGT and OH-GDGT data for Baffin Bay is from this current study (n=13). The other study we reference (Kolling et al., 2020), where we use n=70 data points, is focused on sea ice proxies and only reports HBI and some sterol concentrations. We will make it clear up front in the Materials and Methods as well as throughout the text and figure captions how many samples are being used for which biomarkers. Regarding L283, we are referring to the entire sample set, so we will remove "northern" to avoid any potential confusion. Finally, while yes, we agree that including samples outside the NOW extends the region beyond our focal point, we believe it is necessary to expand the temperature range of our calibration. In this sense, we also use HBI and sterol data from outside the NOW to provide greater context for those biomarkers. As suggested in the following comment, we will add some figures and discussion on the spatial distribution of GDGT and OH-GDGT in Baffin Bay, particularly regarding the NOW.*

It is a pity that the authors have not presented more of the GDGT dataset. For example, the GDGT & OH-GDGT fractional abundance for each of the 13 stations in a table or in a supplementary figure (perhaps similar as to that done for the sterols and HBI concentrations; Fig. S1 - S3). While Figure 6 provides an overview of the entire data set (although I'm still unclear as to whether this is 13 samples or not), I think it would be useful to know whether the GDGT and OH-GDGT distribution varies spatially inside and outside the NOW. Perhaps something similar to the way Spencer-Jones et al. (2021) presented the distribution of archaeal lipid (headgroup)s in different polar water masses. I expect that spatial differences in GDGT & OH-GDGT (core) distributions would not be great in surface sediments, but as you have the data it is a pity not to present it, or at least include a line about any spatial homogeneity or heterogeneity in the results.

*Apologies for the confusion, however, the complete sample datasets will be made available on the PANGAEA online repository upon acceptance of the manuscript. We will add similar spatial figures as provided for the HBI and sterol datasets in the supplemental and add some text in the discussion on the spatial distributions of GDGTs and OH-GDGTs throughout Baffin Bay.*

**Discussion**

Overall, the discussion of GDGTs and OH-GDGTs is clear but I have a few suggestions. The sentence on lines 413 – 415 about the cren isomer reminded me of the analysis in Bale et al. (2019) of the cren' ratio in 58 thaumarchaeotal cultures. They reported that growth temperature has a larger effect than thaumarchaeotal phylogeny on the proportion of the cren isomer.

*We appreciate this insight and will take this into consideration during the revision of our manuscript.*

I found this BGD preprint and discussion https://cp.copernicus.org/preprints/cp-2022-19/ useful when thinking about the relationship between hydroxy-GDGTs and shifting archaeal species composition and/or salinity. Perhaps if it is in print in time, the authors could reference it in relation to the statement on lines 378 – 380 and perhaps in relation to the statement on line 456.

*We were not aware of this preprint, thank you for bringing this to our attention. While we can reference the preprint as it has a DOI, we will hold off until it is published and officially passed the peer review process. If that fall within our revision window, we will certainly reference it.*

A note throughout the manuscript. The authors refer to the cren isomer as a regioisomer, whereas this has been proven not to be the case and is more likely a stereoisomer (Liu et al., 2018; Sinninghe Damsté et al., 2018).

*Thank you and edited.*

One final suggestion I would like to put forward is based on a number of recent articles about decolonization of geosciences, for example Liboiron (2021). Would the authors be prepared to acknowledge the indigenous people that traditionally occupy the regions they sampled? I believe that Inuit Nunangat occupy the West (Canada) and Kalaallit the East (in Greenland). More information about the people in the sample areas and how to acknowledge them can be found at https://native-land.ca/

*We very much appreciate this suggestion as we are always striving to learn and better ourselves for the greater decolonization of geosciences. We had not fully recognized that marine locations could be considered as regions traditionally occupied by indigenous people, which exemplifies the progress that is inherently needed. Acknowledgement and appreciation of the indigenous people has now been added at the end of the paper and will remain in all future relevant publications of ours.*

**Minor edits**

*All minor edits corrected for as suggested below.*

Line 124 – change to "depth"

Line 135 – change to "planktonic"

Line 189 – remove extra )

Line 260 – the second half of this sentence is confusing.

Lines 262 – 263 – This is a confusing sentence as figure 7 only presents temperature correlations and the other variables listed are shown in the supplement. Perhaps easiest here to expand the figure range (e.g., Figs 7, S4 – S6).

Line 265 – change to "their correlation"

Line 269 – is this line missing the word temperature, SST perhaps?

Line 272 – change to either "autumn subsurface temperatures between 60 and 80 m bsl also feature similarly significant correlations" or "autumn subsurface temperature between 60 and 80 m bsl also features similarly significant correlations". I'm not sure which of these you mean but you currently have a mix of both.

Lines 283, 285 – This should be a reference to Figure 8 not 9.

Line 380 – Could you add the actual samples size (i.e. n = x) and temperature range here?

Line 425 – this should be Fig 8c?

---

## Author Comment (AC2)

**Reviewer 2 (anonymous):**

The manuscript by Harning et al. presents the results obtained from 13 surface sediments collected in the largest Arctic polynya (North Water Polynya, NWO). The authors analysed HBIs, sterols, and GDGTs which are used to calculate sea ice- and temperature-related indices. Based on their data, the authors discussed the utility of the paleoproxies and introduced two local calibrations for TEX86-L and RI-OH. Although their attempt sounds reasonable, the dataset is very small with the very narrow temperature range of 2°C and the correlations between indices and temperatures are moderate (R2 <0.5). Nonetheless, the data are valuable since there were no GDGT data from the study area in the global dataset published before. Some issues are listed below, which should be better addressed before the manuscript is accepted.

*We greatly appreciate the reviewer's time and consideration of our manuscript and thank them for a constructive critique that will lead to a stronger paper. Below we address each comment individually.*

**Major comments:**

The authors suggest that all HBIs are derived from sea ice diatoms in Baffin Bay and thus cannot be used to distinguish sea ice and open water conditions. Although they all might be produced by sea ice diatoms, their concentrations and distribution patterns are different. Potentially, they might be derived from different diatom species. Discussion about potential biological sources of individual HBIs can be added based on the literature, although there are no direct information on specific species from Baffin Bay.

*Thank you for the suggestion. We will add some further discussion on the potential sources of HBIs to the discussion section. However, as the reviewer notes, there is no direct information on HBI sources for Baffin Bay, so our discussion will mostly rely on the modern HBI distribution in this study and that of Kolling et al. (2020).*

The TEX86-L calibration was based on 0-90 m water temperatures. But the R2 values are similar to those in 40-90 m water depth as shown in Fig. 7. Although the p values are >0.05 below 90 m water depth, this might be due to insufficient instrumental data. So it will be interesting to show how the calibration based on 0-200 m water temperatures does look like as well.

*Thank you for the suggestion. As can be seen in Figure 2a, temperatures remain relatively isothermal below ~80 m, so it follows that $R^2$ values and calibrations for the deeper depth integrations are similar. However, for the sake of simplicity we choose to focus our discussion on the calibration that features the highest correlation coefficient (i.e., 0-90 m). We will add some text in the discussion to expand upon and clarify this.*

I see that there are no GDGT data previously published in the study area. However, there are HBIs data previously published. I feel that the discussion about HBIs is in general based on 13 samples, not well integrating the previous data from 70 sites. These data are not even incorporated in Fig. S1 to S3. It is not clear what might be the reason.

*We have now added the previously published dataset from Kolling et al. (2020) to our supplementary figures for comparison and will expand our discussion to better integrate their dataset.*

**Other comments:**

Line 177: an Agilent DB-1MS GC column (60 m x 250 μm x 250 μm)? Is the column information correct?

*Apologies for the typo regarding the film thickness – it is in fact 0.25 um. This has now been corrected for both the DB1 and DB5 columns.*

Line 182-185: Concerning to the response factors for HBIs quantifications, it is not clear how the approach used in this paper is comparable to that used in the paper by Belt et al., 2012.

*Apologies for any confusion. During our analyses we did not have access to the internal standards used by Belt et al. (2012, e.g., 7-HND and 9-OHD, Belt et al., 2012), and therefore 3-methylheneicosane as our internal standard for the aliphatic hydrocarbon fraction. To account for the varying response factor of our internal standard and those of Belt et al. (2012), and make our datasets comparable with other HBI studies, including Kolling et al. (2020), we obtained the 7-HND and 9-OHD standards and ran a 5-point external dilution series along with 3-methylheneicosane. We then calculated sample HBI concentrations using the response factor of our internal standard (3-methylheneicosane) after correction for the difference in response factors of 7-HND and 9-OHD. We have now explained this in more detail in the text and have also added figures for our external HBI and sterol dilution series to the supplement.*

Line 194-196: Similarly, concerning to the response factors for sterols quantifications, it is not clear why cholesterol is used instead of target sterols directly. The standard samples for ß-sitosterol, brassicasterol, and campesterol are available in the markets, except for dinosterol.

*While we are aware of these standards' availability, we used cholesterol as an external standard as the study we directly compare with (Kolling et al., 2020) also used cholesterol.*

HBI III and HBI IV: It would be good to show HBIs chemical structures as a supplementary figure.

*Thank you for the suggestions, we will add HBI structures to the supplement.*

Line 236-237 & Fig. 4a: Looking at Fig. 4A, the standard deviations of mean concentrations of all HBI compounds appear to be overlapping. To better demonstrate the difference, some additional statistics should be done.

*Agreed, we will now add t-test and p-value results to better support the statistical differences or not. Description of these methods is now also added to the Methods and Materials section to more clearly lay out how we statistically assess our datasets.*

Line 240-246 & Fig. 4c: The sample number can be added. In addition, some statistical analyses should be done to better demonstrate whether the datasets between NOW and non-NOW are different. Although it is written in the text like "Although the standard deviations of

mean dinosterol and campesterol concentrations overlaps between NOW and non-NOW sites, the standard deviations of ß-sitosterol and brassicasterol for the two regions is statistically different (Fig. 4c and S2).", Figure 4c rather shows that ß-sitosteroal and dinosterol are different between NOW and non-NOW while the standard deviations of meanbrassicasterol and campesterol overlap.

*We will add the sample numbers and perform t-tests to better demonstrate the statistical differences between the NOW and non-NOW datasets. The latter issue was a typo. The reviewer correctly notes the proper biomarkers that we deemed different or not, and this will be corrected.*

Line 249: The balance factors were obtained based on a combination of previous and current datasets. However, later on, the major conclusions related to the PIP25 indices were based on the current study (i.e. n=13). How are the balance factors if they are calculated only based on the current study? Are the resulting PIP25 values similar?

*Considering that the balance factor is derived from mean IP25 to mean "phytoplankton biomarker" concentrations, and that the concentrations of these biomarkers are relatively similar between the two studies, the c factors are similar whether one relies solely on Kolling et al. (2020) or this study. However, to be more inclusive and take advantage of a larger dataset, we merged the two data sets to use all the local available data.*

Line 259: "….the standard deviation of mean values between these regions is not statistically different (Fig. 4e and S3)." – What kind of statistics were done? The statistical results were not shown to compare both NOW and non-NOW datasets.

*Sorry for any confusion. We used the range of standard deviations as a test for significant differences between NOW and non-NOW datasets. We have now conducted t-tests to ascertain the statistical differences more robustly and will amend any changes to the text accordingly.*

Line 266-280: The authors show the R2 values but it is not clear on how many samples these are based. Please provide the number of samples.

*Apologies for any confusion – the GDGT data is from all 13 samples presented in this study. This has been clarified in the Methods and Materials and sample numbers added to all figure captions.*

Line 281-286: There is no Fig. 9a and 9b.

*Thank you for catching this typo. It should read Fig. 8 and has now been corrected.*

Line 320-322: In the Kolling's dataset, HBI III is correlated with dinosterol and brassicasterol which is not observed in the current study. What would be the reason?

*Per Figure 5b, there are indeed correlations, albeit weak, between HBI III and dinosterol ($R^2$=0.17) and brassicasterol ($R^2$=0.21) in the Kolling dataset. Similarly weak (but insignificant) correlations are also observed between HBI III and dinosterol ($R^2$=0.30) and brassicasterol ($R^2$=0.25) in our dataset suggesting consistency between the two studies. However, ours are not plotted in Figure 5a as the p value was >0.05, which may be partially attributed to the small size of the dataset (n=13) compared to Kolling (n=70). In any case, we do not interpret these*

*weak regression coefficients as indicative of important correlations. We will make this clearer in the main text.*

Line 325-326 & Fig. 4: There are also some differences in sterols between two datasets. Is there any possibility that this is due to the different quantification methods applied?

*The only difference in the analysis of the two studies sterol datasets is our external dilution series that corrects for the different response factors of internal standards. Our lab's quantification protocol is very thorough and should account for these different internal standards. Therefore, we are not sure why there are some differences in sterol concentrations between the two datasets beyond what we originally posited as a possible geographic control in the main text.*

Line 345-347: It is somewhat confusing to see the difference for PBIP25 and PDIP25 between two datasets. How does it look like if the data from the sites in front of fjords in the Kolling's dataset are removed? If so, is the difference less then?

*Per the reviewer's suggestion, we tested whether removing sites closer to the fjords in the Kolling et al. (2020) dataset would bring the PIP indices into closer alignment with the samples from our study. Unfortunately, this did not substantially change the results. However, upon statistical analysis of the NOW and non-NOW sites using t-tests following this reviewer's earlier suggestion, none of the PIP mean values in the NOW are statistically different from mean values outside the NOW. Therefore, this section will be amended, and the differences as noted between the two datasets by the reviewer in terms of PBIP25 and PDIP25 may simply be the result of the different number and distribution of samples between the two studies. However, the mean value differences should not be viewed as statistically different.*

Line 401: Besides the regression analysis, other statistical analyses, such as PCA and RDA would be helpful to better illustrate the impact of the main environmental factors on the GDGT distributions.

*While we appreciate the suggestion by the reviewer, we originally opted not to conduct this analysis to the large number of environmental variables across different seasons and depth integrations. In our opinion, plotting all these variables, along with the 13 samples would produce plots that are too cluttered and difficult to read as well as add more figures that may overwhelm the reader. In addition, we believed that the illustration of our sample and environmental variable relationships would not reveal anything that is not already apparent in our figures. To be open, we show below a PCA analysis for the surface (25 m depth) using annual temperature, salinity, DO and nitrate. As can be seen in Figure 7, annual SST plots closely with the RI-OH index at 25 m depth, and not the other GDGT indices. The RI-OH index also plots closely with DO, as can be seen in Figure S5 at this depth. Therefore, we respectfully intend to leave the regression analyses as is for our GDGT and OH-GDGT datasets.*

[Figure]

Fig. 5: It is a little bit confusing to see the correlations between the same compounds. It is obvious that they have the value of 1. It would be better to remove them.

*Agreed, and will remove. Thank you.*

Fig. S1, S2, and S3: The color bar scale is not visible.

*We apologize for any difficulties and have adjusted the scale bars to be more visible.*

Table: It would be beneficial for other researchers to present data of individual HBIs, sterols and GDGTs as an Exel file or tables in Supplementary Information, although they can be deposited in a website later on.

*We absolutely agree and plan on submitting our datasets to the PANGAEA online repository upon acceptance of our manuscript. However, we can also include the data as supplemental material for easier access to the reader.*

---

## Author Comment (AC3)

*Reviewer 3 (Sofia Ribeiro):*

Harning and co-authors present a study of biomarker distribution in surface sediment samples from Baffin Bay, with focus on identifying biomarker signatures for the North Water Polynya that may inform paleo-environmental reconstructions in the region. This is both a well-justified and significant effort. On the one hand, the North Water is a globally important ecosystem that is particularly vulnerable to climate change. On the other hand, there is considerable uncertainty in the behavior, source producers (HBIs and sterols), and fidelity of biomarker proxies and thus local and regional data are essential to refine their usability.

The study includes a broad range of biomarkers and this is, in my opinion, the main strength of the work. It is also very well-written, and the figures are generally well prepared and adequate (see below for some suggestions for improvement).

My comments will focus on the HBIs and sterol work, as GDGTs are outside my realm of expertise.

While the study provides some useful insights (adding, for example, to the discussion on whether HBI III and HBI IV might in fact be produced by sympagic taxa), I found some of the data analyses and conclusions problematic and cannot recommend publication of the paper in its present form. I encourage the authors to consider a few points and revise the manuscript accordingly.

*We greatly appreciate Sofia Ribeiro's time and consideration of our manuscript and thank the reviewer for a constructive critique that will lead to a stronger paper. Below we address each comment individually.*

**General comments:**

- Limited number of samples and novelty

The study includes a surprisingly low number of samples (n=13, n=9 analyzed for bulk geochemistry) to represent such a large area. I found there are some overstatements throughout the manuscript that should be corrected having in mind the small sample size, and what indeed it adds in terms of novelty (or not) to the large Kolling et al. 2020 study. For example, in the Introduction it is mentioned that samples were collected in 2008 and 2017 but the total number is not given. From the table I assume 2008 (n=3), 2017 (n=10). It is also written in the introduction that biomarker proxies were assessed against modern instrumental data but this was only presented for GDGTs. Another overstatement, although minor, in the discussion (line 299) is saying that "several additional sterols" were added compared to the work of Kolling et al. 2020 when in fact only two more were analysed.

*We fully acknowledge the limitation of our dataset and have tried to highlight this where relevant in the manuscript. Part of this study's novelty is that we include a broader assessment of biomarkers than traditional studies, as well as take advantage of large, previously published datasets (e.g., Kolling et al., 2020). For the latter, the authors presented HBI and sterol datasets to explore the utility of sea ice proxies in Baffin Bay, but not regarding the NOW specifically. We will edit the main text to remove and/or reduce any overstatements as pointed out here and highlight the novelty where needed.*

- Lack of information on surface sediment samples

In order to be able to assess the new findings, it is important that the authors provide additional information on the samples. Table 1 can be expanded to include at least "year of collection" (might not be obvious to all from the cruise code), "coring device", and data on bulk geochemistry. Figure 3 shows 13C and C/N data for the 9 samples, but it is not clear which ones these are. Also, TOC values should be given in the table, as these are important when considering biomarker concentrations (see comment below also). Is there any age control on the surface samples? Were any of these cores analysed for 137Cs/210Pb? This would give some confidence at least that they might indeed represent recent deposition.

*We apologize for any information that we did not include that may be important for the reader. In the revised manuscript, we will add the year of collection, coring device, and indicate which samples were analyzed for bulk geochemistry to Table 1. Data for bulk geochemistry will be submitted along with the biomarker data to the PANGAEA online repository. Unfortunately, we do not have any quantitative age control on the surface samples, so we restrict ourselves to providing the thickness of the sample interval in centimeters. However, we do note that these samples all contain modern sediment as stained living foraminifera were present.*

- Comparison with Kolling et al 2020

I found some of the comparisons with the Kolling et al. dataset quite confusing. In the materials and methods, it is written that the new dataset (n=13) will be compared with a subset of samples (n=70) from Kolling et al and that samples collected within fjords and bays will be excluded. However, later in the discussion, it is argued that the difference in brassicasterol and dinosterol trends between the two are likely due to the fact that some of the sites are in the vicinity of large fjords. If there is uncertainty whether some samples might be skewing the response, the authors could run the comparison with a different subset and evaluate if this is the case. Given that the Kolling et al dataset includes many more samples, the ranges of water depths, sedimentation rates, and likely sediment composition are likely larger than for the n=13 dataset. Simply comparing biomarker concentrations per volume of sediment across the two datasets and the now vs. non-now sites is not adequate, in my opinion.

*Apologies for any confusion. We removed any sites from Kolling et al. (2020) from immediately within the fjords and bays. The sample sites that we mention regarding brassicasterol and dinosterol trends are not within the fjords/bays but located proximally within the main region of Baffin Bay. As suggested by reviewer 2, we did test whether removing these sites brings the two study's datasets into closer agreement, however, that was not the case. We will investigate whether these differences may be the result of varying TOC content or not.*

- Influence of TOC contents on the biomarker signals

Given that the NOW is a highly productive area, one can expect that TOC values for the NOW sites will be generally higher than for the non-NOW sites. It has been recommended, and is common practice in paleo sea ice reconstructions using HBIs, to normalize the data by TOC. This way, we account for down-core changes in sediment composition. The same would be important for a dataset like this one, where large changes in sediment composition and organic matter content can be expected across the region. I strongly recommend that the authors plot all biomarker data in ng.gTOC-1 and revise the discussion and conclusions accordingly. Figures S1 and S2 might also show quite different spatial trends if TOC values are accounted for.

*Thank you for the recommendation. We had originally opted not to normalize biomarker concentrations against TOC as not all samples were analyzed for bulk geochemistry due to insufficient sample, and for the ones that had been, it did not significantly alter the results. By focusing on concentrations, we could have a slightly larger dataset to work with. In any case, we do see the value of presenting this data, and plan to include it in the supplementary files and incorporate it into the discussion of the main text.*

- Recommendations for paleoenvironmental reconstructions

This study highlights the complexity of biomarker signals in the highly dynamic Baffin Bay region, and our limited knowledge of their mechanistic behavior and applicability. Given my previous comments, and the uncertainty linked to comparing NOW vs. non-NOW sites based on concentrations of biomarkers per volume of sediment without accounting for sediment composition and in the absence of any form of age control, I cannot agree with the recommendation of proposing one type of biomarker (sterols) as a "more appropriate tool" (rather than HBIs) to characterize the NOW in the recent past. On the contrary, I think this study is a perfect example of why we need to pursue a multiproxy approach and not rely on single proxy lines of evidence. I would like to mention here that previous Holocene records from the NOW have mostly followed a multiproxy approach including microfossils, biomarkers, and biogeochemical proxies and I would be concerned if the community, based on this study, would go ahead and attempt to reconstruct the NOW based on sterols alone.

*We appreciate this comment and the reviewer's concern and will clarify in the text that a multi proxy approach is always recommended. Given our statistical analyses, the sterols indeed have significantly different concentrations in the NOW versus sites outside the NOW. This remains true whether they are presented as concentrations or normalized to TOC. While we agree that the analysis of other supporting proxies is needed to help elucidate why and how the NOW formed in the past (e.g., sea ice and temperature), as they are indeed responsible for its complex development, our results suggest that sterols will be useful to pinpoint when the NOW developed as it is today.*

**Detailed comments:**

Lines 7 and 38 - Greenlandic Inuit is not a language. The correct term is West Greenlandic or Kalaallisut.

*Thank you for highlighting this. We realized this ignorant mistake after submission, which has been now corrected. We also now include the local Eastern Canadian Inuit name, to be more inclusive and acknowledge all the traditional inhabitants of this paper's study area.*

Lines 33,34 – The instability of the NOW (e.g. Ribeiro et al. 2021) has been shown by a combination of multiple proxies, including lipid biomarkers, microfossils and bulk biogeochemistry (not just lipid biomarkers).

*Absolutely, amended in the text.*

Lines 44-45 – human occupation timelines are incomplete and outdated, please correct. See:

- Ribeiro et al. 2021 Fig 5 (already cited in this manuscript) and references therein, mainly: 1) Raghavan, M. et al. The genetic prehistory of the New World Arctic. Science **345**, 1255832

(2014). And **2)** Grønnow, B. & Sørensen, M. Palaeo-Eskimo migrations into Greenland: The Canadian Connection. In Dynamics of Northern Societies. Proceedings of the SILA/NABO Conference on Arctic and North Atlantic Archaeology, Copenhagen (eds Arneborg, J. & Grønnow, B.) 59–74 (National Museum, Studies in Archaeology & History, 2006).

*Thank you for bringing this to our attention. The timing and references have now been updated accordingly.*

Line 47 – This study should be mentioned here as well: Vincent, R. F. A study of the North Water Polynya ice arch using four decades of satellite data. Sci. Rep. **9**, 20278 (2019).

*Done, and thank you for the suggestion.*

Line 146 – specify coring device in the table per sample

*We agree this is important information to include and have now added it to Table 1.*

Line 149 – here, add any information on age control for the core tops if possible.

*Unfortunately, we do not have any quantitative age control. We have added a clearer assumption of their recent ages to the Materials and Methods section 4.1.*

Line 226 – did you mean to write "shoulder season months"? I am not familiar with this expression.

*Yes, shoulder season refers to autumn and spring months. This has been clarified in the text.*

Line 229 – TOC data should be added here. Specify in the table which of the 9 samples were analysed for bulk geochemistry.

*Thank you for the suggestion, both will be added to the manuscript.*

Line 307-308 – One could argue for the opposite, given that polynyas are characterized by intense sea ice formation, and the polynya area is under the influence of Arctic sea ice export. I suggest revising this section.

*Apologies if we do not understand this comment fully, but we do describe these possibilities as suggested by the reviewer in the sentences immediately following L307-308.*

Line 327 – Would be useful to add more information here on other potential sources for campesterol and b-sitosterol (besides marine diatoms). Also note that the Detleft et al 2021 study is in a fjord setting, while the datasets in this study excludes such settings. I don't completely follow the reasoning that correlation (of sterols) is supportive of a common source.

*Apologies if our reasoning was not clear. We are aware that the Detlef study is from a fjord but wanted to mention it as it is the only other study that has reported campesterol and b-sitosterol from the greater Baffin Bay region. As we discuss in the Section 3 (Background on biomarkers) and in the sentences following L327 noted here by the reviewer, there are other potential sources of these sterols, such as vascular plants (Huang and Meinschein, 1976). While vascular*

*plants are commonly taken as the primary source for these sterols, we note that terrestrial biomass in this region of the Arctic is low (Gould et al., 2003), and without major river networks to transport the minimal biomass to the ocean, we assume that the contribution of these sterols from vascular plants is low compared to those produced by marine diatoms. Of course, this is only an assumption, but supported by the correlation of sterols in our surface sediment samples. Further detailed work on the production and transport of these sterols from algal and vascular plant samples in and around Baffin Bay will undoubtedly benefit paleoceanographic interpretations in future studies. We additionally mention this final point in the concluding paragraph on our suggestions for future studies.*

Line 350 – replace "all" with "partly" – biomarkers may partly originate from sea ice diatoms

*Done.*

Lines 365-366 – please revise this conclusion. Sterols alone are very unlikely to help us characterize the presence/absence of the NOW in the recent geological past, and other tools are available, such as "true" open water indicators.

*Similar to the earlier comment by the reviewer, we will edit the text to clarify that multi proxy approaches are always favored to better assess the complex variables that lead to the NOW's past development. However, based on our statistical analyses (t-tests), sterols are the only biomarkers that are significantly different between the NOW and sites outside the NOW. While the other biomarkers in our study, as well as other proxies such as microfossil assemblages, provide important context for changes in the local paleoceanography, we believe based on our data that sterols will indeed be key to helping reconstruct the presence of the NOW.*

Lines 472-473 – It is important to verify if this holds when accounting for TOC contents.

*We note that importance of accounting for TOC as previously suggested by the reviewer, but the correlations between individual HBIs and sterols from the same sample would not change whether we normalize biomarker concentrations to sample mass or TOC. Therefore, this still holds true.*

Line 479 – I suggest some caution here since sterols are not unequivocal open water indicators.

*As we discussed in our previous response, our data and statistical analyses support this line of reasoning. While it is indeed true that sterols are not unequivocal open water indicators, our data suggest that sterols here may be dominantly sourced from open water primary producers in the area, rather than from other sources such as sea ice diatoms or vascular plants. This aspect will be further clarified in the main text.*

Figure 4 – I suggest replotting with TOC normalized values

*Due to the smaller number of samples that contain TOC information (n=9), we have added these suggested plots and comparisons with Kolling et al. (2020) to the supplement, rather than include in the main text.*

Figure 5 – It took me a while to make sense of this figure. I suggest ordering the biomarkers in the same way as Fig 4 so they are easily comparable. IP25, HBI II, HBI III, HBI IV, Dinosterol,

Brassicasterol, Campesterol, b-sitosterol. Also specify if the plotted data in b) are all Kolling et al or a sub-set as indicated in the text? Sample sizes (n=) should be given for all figures.

*Thank you for the good suggestions. Both will be amended accordingly.*